# Quantitative H$_2$S-mediated protein sulfhydration reveals metabolic reprogramming during the integrated stress response

Xing-Huang Gao[1], Dawid Krokowski[1], Bo-Jhih Guan[1†], Ilya Bederman[2†], Mithu Majumder[3], Marc Parisien[4], Luda Diatchenko[4], Omer Kabil[5], Belinda Willard[6], Ruma Banerjee[5], Benlian Wang[7,8], Gurkan Bebek[7,8], Charles R. Evans[9], Paul L. Fox[10], Stanton L. Gerson[11], Charles L. Hoppel[3], Ming Liu[9], Peter Arvan[9], Maria Hatzoglou[1*]

[1]Department of Genetics and Genome Sciences, Case Western Reserve University, Cleveland, United States; [2]Department of Pediatrics, Case Western Reserve University, Cleveland, United States; [3]Department of Pharmacology, Case Western Reserve University, Cleveland, United States; [4]Alan Edwards Centre for Research on Pain, McGill University, Montreal, Canada; [5]Department of Biological Chemistry, University of Michigan Medical School, Ann Arbor, United States; [6]Mass Spectrometry Laboratory for Protein Sequencing, Lerner Research Institute, Cleveland Clinic, Cleveland, United States; [7]Center for Proteomics and Bioinformatics, School of Medicine, Case Western Reserve University, Cleveland, United States; [8]Center for Synchotron Biosciences, School of Medicine, Case Western Reserve University, Cleveland, United States; [9]Department of Internal Medicine, Division of Metabolism, Endocrinology & Diabetes, University of Michigan Medical School, Ann Arbor, United States; [10]Department of Cellular and Molecular Medicine, Lerner Research Institute, Cleveland Clinic, Cleveland, United States; [11]Department of Medicine, Division of Hematology/Oncology, School of Medicine, Case Western Reserve University, Cleveland, United States

*For correspondence: mxh8@case.edu

†These authors contributed equally to this work

Competing interests: The authors declare that no competing interests exist.

**Abstract** The sulfhydration of cysteine residues in proteins is an important mechanism involved in diverse biological processes. We have developed a proteomics approach to quantitatively profile the changes of sulfhydrated cysteines in biological systems. Bioinformatics analysis revealed that sulfhydrated cysteines are part of a wide range of biological functions. In pancreatic $\beta$ cells exposed to endoplasmic reticulum (ER) stress, elevated H$_2$S promotes the sulfhydration of enzymes in energy metabolism and stimulates glycolytic flux. We propose that transcriptional and translational reprogramming by the integrated stress response (ISR) in pancreatic $\beta$ cells is coupled to metabolic alternations triggered by sulfhydration of key enzymes in intermediary metabolism.

## Introduction

Posttranslational modification is a fundamental mechanism in the regulation of structure and function of proteins. The covalent modification of specific amino acid residues influences diverse biological processes and cell physiology across species. Reactive cysteine residues in proteins have high nucleophilicity and low pKa values and serve as a major target for oxidative modifications, which can

**eLife digest** Proteins play essential roles in almost every aspect of a cell's life, and also contribute to the structure and function of body tissues and organs. Cells and tissues adapt to their continuously changing environments by regulating the activity of their proteins. For example, proteins that are not fully active immediately after they are built instead require further 'posttranslational' modifications to become active.

Amino acids are the building blocks of proteins, and cysteine amino acids are frequent sites of posttranslational modifications because they are particularly chemically reactive. Under certain conditions inside the cell, the sulfur atom in a cysteine can bond with chemical group containing a second sulfur atom plus a hydrogen atom. This process, which is known as sulfhydration, can be triggered by the presence of the gas molecule, hydrogen sulfide ($H_2S$). The levels of hydrogen sulfide are highly regulated in the body, and it has been suggested to play a role in aging, environmental stress and many diseases. However, it is not clear whether sulfhydration plays a major role in disease conditions by modifying protein activity.

Efforts to address this question have been limited by a lack of methods that can measure the extent of sulfhydration of proteins. However, Gao et al. have now devised such a method. The approach takes steps to avoid false-positive and false-negative results, and can identify changes in the sulfhydration of cysteines across the entire complement of proteins produced by a cell, tissue or organ.

Gao et al. then used this new method to show that a master regulator of transcription (i.e. a protein that regulates the expression of many genes) causes large-scale changes in cysteine sulfhydration. These large-scale changes resulted in the reprogramming of the cell's energy metabolism, and further experiments showed that hydrogen sulfide accumulation influences sulfhydration, protein activity and signaling pathways. The development of this new method may now lead to additional discoveries into the role of hydrogen sulfide as a signaling molecule.

vary depending on the subcellular environment, including the type and intensity of intracellular or environmental cues. Oxidative environments cause different post-translational cysteine modifications, including disulfide bond formation (-S-S-), sulfenylation (-S-OH), nitrosylation (-S-NO), glutathionylation (-S-SG), and sulfhydration (-S-SH) (also called persulfidation) (*Finkel, 2012*; *Mishanina et al., 2015*). In the latter, an oxidized cysteine residue included glutathionylated, sulfenylated, and nitrosylated on a protein reacts with the sulfide anion to form a cysteine persulfide. The reversible nature of this modification provides a mechanism to fine tune biological processes in different cellular redox states.

Sulfhydration coordinates with other post-translational protein modifications such as phosphorylation and nitrosylation to regulate cellular functions (*Altaany et al., 2014*; *Sen et al., 2012*). Despite great progress in bioinformatics and advanced mass spectroscopic (MS) techniques, identification of different cysteine-based protein modifications has been slow compared to other post-translational modifications. In the case of sulfhydration, a small number of proteins have been identified, among them the glycolytic enzyme glyceraldehyde phosphate dehydrogenase, GAPDH (*Mustafa et al., 2009*). Sulfhydrated GAPDH at Cys[150] exhibits an increase in its catalytic activity, in contrast to the inhibitory effects of nitrosylation or glutathionylation of the same cysteine residue (*Mustafa et al., 2009*; *Paul and Snyder, 2012*). The biological significance of the Cys[150] modification by $H_2S$ is not well-studied, but $H_2S$ could serve as a biological switch for protein function acting via oxidative modification of specific cysteine residues in response to redox homeostasis (*Paul and Snyder, 2012*). Understanding the physiological significance of protein sulfhydration requires the development of genome-wide innovative experimental approaches. Current methodologies based on the modified biotin switch technique do not allow detection of a broad spectrum of sulfhydrated proteins (*Finkel, 2012*). Guided by a previously reported strategy (*Sen et al., 2012*), we developed an experimental approach that allowed us to quantitatively evaluate the sulfhydrated proteome and the physiological consequences of $H_2S$ synthesis during chronic ER stress. The new methodology allows a quantitative, close-up view of the integrated cellular response to environmental and intracellular cues and is pertinent to our understanding of human disease development.

## Results

The ER is an organelle involved in the synthesis of proteins followed by various modifications. Disruption of this process results in the accumulation of misfolded proteins, causing ER stress (*Tabas and Ron, 2011*; *Walter and Ron, 2011*), which is associated with development of many diseases ranging from metabolic dysfunction to neurodegeneration (*Hetz, 2012*). ER stress induces transcriptional, translational, and metabolic reprogramming, all of which are interconnected through the transcription factor ATF4. ATF4 increases expression of genes promoting adaptation to stress via their protein products. One such gene is the $H_2S$-producing enzyme, cystathionine gamma-lyase (CTH) or γ-cystathionase , previously shown to be involved in the signaling pathway that negatively regulates the activity of the protein tyrosine phosphatase 1B (PTP1B) via sulfhydration (*Krishnan et al., 2011*). We therefore hypothesized that low or even modest levels of reactive oxygen species (ROS) during ER stress may reprogram cellular metabolism via $H_2S$-mediated protein sulfhydration (*Figure 1A*).

We have previously shown that the insulin-producing mouse pancreatic β cells MIN6, known for their high metabolic activity, are very susceptible to ER stress (*Guan et al., 2014*; *Krokowski et al., 2013*). We tested whether or not MIN6 cells expressed the essential components of $H_2S$ synthesis and protein sulfhydration in response to ER stress induced by thapsigargin (Tg, *Figure 1*). Upon Tg treatment, MIN6 cells exhibited higher levels of intracellular ROS, decreased GSH/GSSG (reduced/oxidized glutathione ratios), and increased CTH protein levels via transcriptional activation (*Figure 1B–D*, *Figure 1—figure supplement 1* and *2A*). This transcriptional reprogramming was also seen in mouse and human islets subjected to physiological or pharmacologically induced ER stress (*Figure 1D*, *Figure 1—figure supplement 2B, C* and *3*). In agreement with increased CTH expression, $H_2S$ levels increased during the chronic phase of the stress response (*Figure 1E*). Coordinated induction of expression of the gene encoding the glutamate/cystine exchanger, Slc7a11, also was increased (*Figure 1D*), and this induction was associated with increased glutamate/cystine flux (*Figure 1—figure supplement 4*). Slc7a11 mediates the exchange of oxidized extracellular cystine with intracellular glutamate. Once cystine is imported into the cells it gets reduced to cysteine, and then serves as a substrate for GSH and $H_2S$ synthesis. These data support the idea that both increased uptake of the CTH substrate and increased levels of CTH contribute to increased $H_2S$ levels in cells under ER stress.

The functional significance of increased $H_2S$ synthesis was shown by measuring the catalytic activity of GAPDH (*Mustafa et al., 2009*), which gradually increased in response to elevated $H_2S$ production in MIN6 cells during ER stress (*Figure 1F*). This increase in activity was independent of GAPDH protein levels (*Figure 1F*). As noted above, GAPDH has the unusual feature of being catalytically inactive when $Cys^{150}$ is oxidatively modified, except when it undergoes sulfhydration which restores/increases its activity (*Mustafa et al., 2009*). We therefore tested the protective effects of $Cys^{150}$ sulfhydration by $H_2S$ on its catalytic activity in the presence of $H_2O_2$-induced oxidation. Recombinant GAPDH was incubated with $H_2O_2$ in the presence or absence of the $H_2S$ donor, NaHS. The inhibition of GAPDH activity by $H_2O_2$ was significantly reversed by $H_2S$ treatment (*Figure 1G*). Furthermore, incubation of purified GAPDH with oxidized glutathione (GSSG) resulted in formation of inactive glutathionylated GAPDH (*Gao et al., 2010*), which was significantly rescued by treatment with $H_2S$ as well as DTT reduction (*Figure 1H*). HPLC-MS confirmed that recombinant GAPDH exposed to NaHS or GSSG was modified predominantly at $Cys^{150}$ (*Figure 1—figure supplement 5*, *Figure 2—figure supplement 7B, C*). These data confirm that $H_2S$ is a positive regulator of GAPDH activity. Increased GAPDH activity is directly linked to $H_2S$ production, as shown by the loss of induction (*Figure 1I*) upon treatment with the CTH inhibitor, propargylglycine (PAG). PAG inhibits $H_2S$ synthesis and therefore is expected to decrease $Cys^{150}$ modification. Taken together, these results indicate that regulation of $H_2S$ synthesis during ER stress might regulate the catalytic activity of other metabolic pathway proteins. The latter is raising the possibility that the ATF4-mediated sulfhydration of proteins is part of the integrated stress response (ISR), and has regulatory effects on cellular metabolism.

ATF4 increases gene expression of CTH (*Dickhout et al., 2012*), the cystine transporter Slc7a11, as well as the ROS-producing enzyme Ero1α (*Han et al., 2013*; *Tabas and Ron, 2011*). We hypothesized that during ER stress, this network of ATF4 target genes promotes protein sulfhydration (*Figure 2A*). Knockdown of ATF4 in MIN6 cells during Tg-induced ER stress caused inhibition of $H_2S$

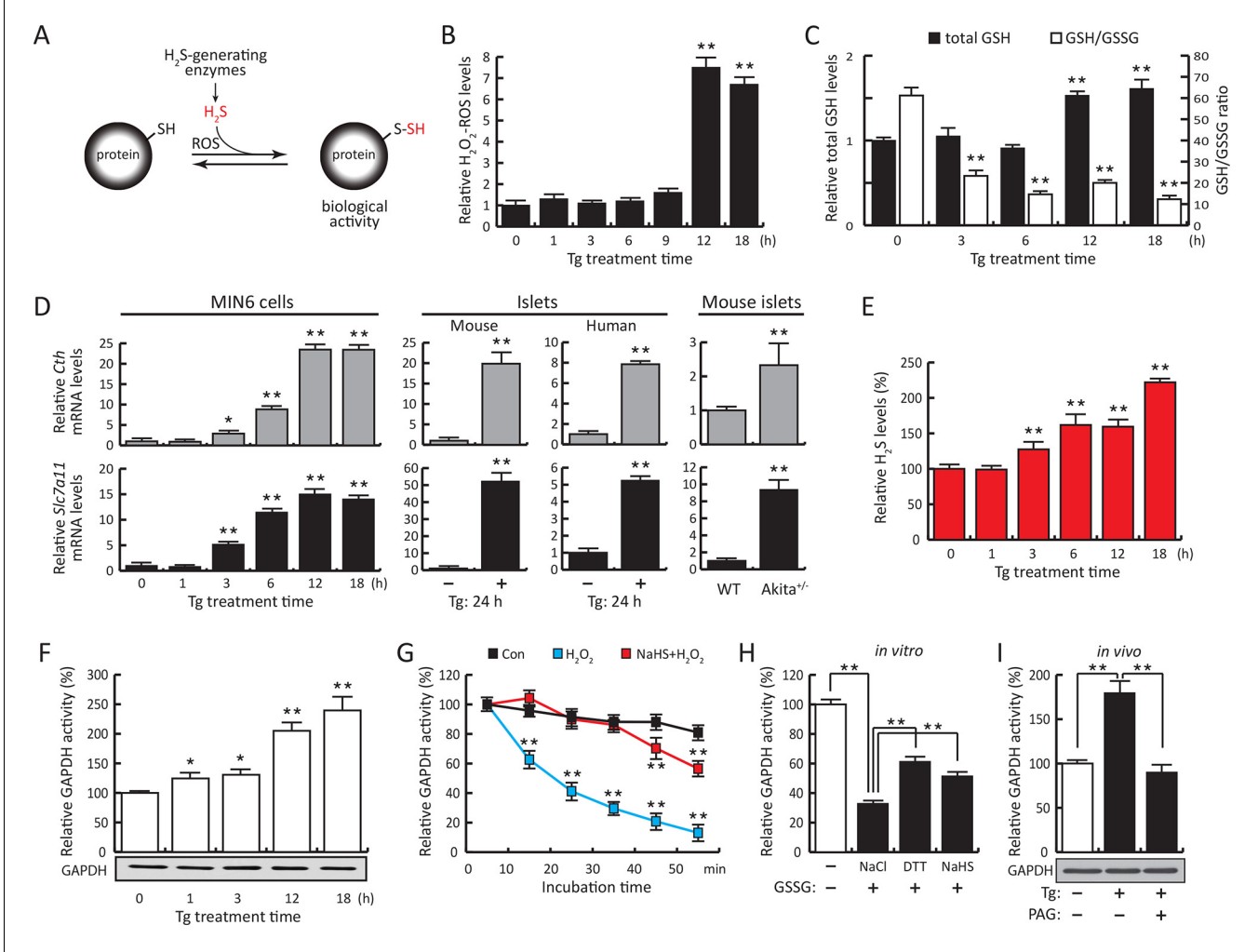

**Figure 1.** ER stress induces protein sulfhydration, a reversible cysteine-based post-translational modification. (A) Schematic overview of protein sulfhydration, which requires synthesis of $H_2S$ and low ROS levels. Pancreatic β cells (MIN6) were treated with thapsigargin (Tg) for the indicated times, and the cellular levels of ROS (B), total levels of GSH and GSH/GSSG ratios (C) and $H_2S$ levels (E) were evaluated. (D) RT-qPCR analysis of the mRNA levels for the $H_2S$-producing enzyme CTH and the cystine/glutamate exchanger Slc7a11 in MIN6 cells treated with Tg or pancreatic islets as indicated. (F) Evaluation of GAPDH activity in cell extracts from MIN6 cells treated with Tg at the indicated times (top), and GAPDH protein levels by Western blot analysis (bottom). (G). Time-dependent measurements of human recombinant GAPDH activities after exposure to $H_2O_2$ (50 μM, blue) or $H_2O_2$ together with the $H_2S$ donor, NaHS (50 μM, red). (H) In vitro evaluation of the reversal of the inhibitory effect of glutathionylation on the activity of recombinant GAPDH treated for 15 min with either NaHS (20 mM), DTT (reduced dithiothreitol, 20 mM), or NaCl (20 mM). (I). Evaluation of GAPDH activities in MIN6 cell extracts either untreated or treated with Tg (18 hr) with or without the CTH inhibitor, PAG (3 mM) (top). PAG was included for the last 3.5 hr of Tg-treatment. GAPDH protein levels were evaluated by Western blotting (bottom). All quantifications are presented as mean ± S.E.M. of three independent experiments. CTH: γ-cystathionase; ER: endoplasmic reticulum; PAG: propargylglycine; ROS: reactive oxygen species.

The following figure supplements are available for figure 1:

**Figure supplement 1.** ER stress induces the levels of the $H_2S$-producing enzyme CTH but not CBS.

**Figure supplement 2.** Regulation of gene expression in MIN6 cells, human, and mouse islets in response to ER stress.

**Figure supplement 3.** Activation of the integrated stress response leads to increased expression of CTH in wild-type mouse islets treated with Tg.

**Figure supplement 4.** Glutamate uptake in MIN6 cells treated with Tg at the indicated times.

**Figure supplement 5.** Analysis of S-glutathionylated GAPDH by LC-MS/MS.

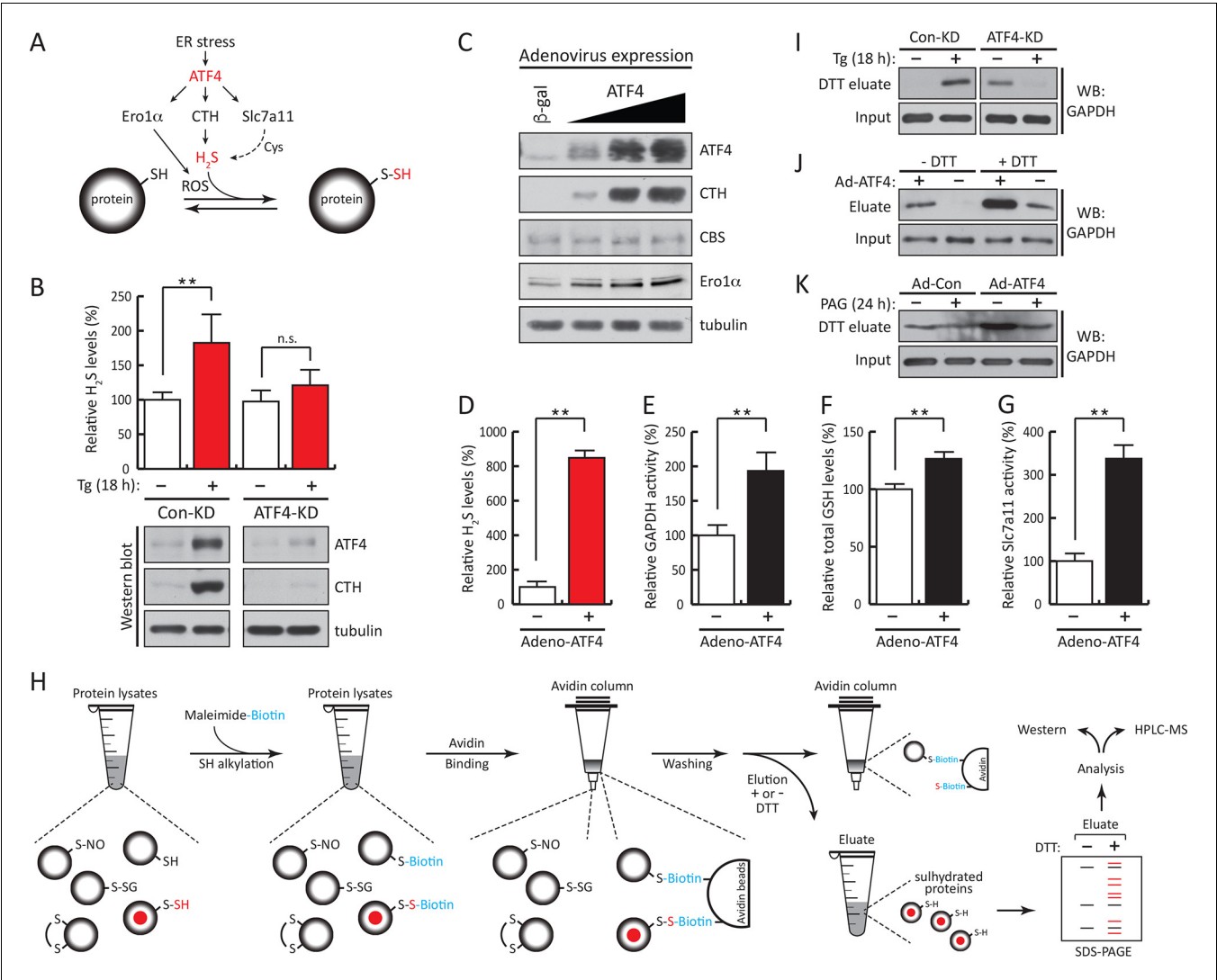

**Figure 2.** ATF4-mediated transcriptional reprogramming during ER stress increases expression of a gene cohort involved in H₂S synthesis and protein sulfhydration. (**A**) Schematic representation of the ATF4-induced cohort of genes leading to sulfhydration of proteins during ER stress. (**B**) Evaluation of intracellular H₂S levels (top) or ATF4 and CTH protein levels (bottom) in MIN6 cells infected with either control shRNA or shRNA against ATF4. Cells were untreated or treated with Tg for 18 hr. (**C**) Western blot analysis for the indicated proteins in MIN6 cells infected (for 48 hr) with either adenovirus mediated lacZ-expression as control, or ATF4-expression, at increasing viral particle concentrations. MIN6 cells infected with either control adenovirus or ATF4-expressing adenovirus were used to measure (**D**) H₂S levels (**E**) GAPDH relative activities, (**F**) GSH levels, and (**G**) Glutamate (Glu) uptake by the cystine/glutamate exchanger. (**H**) Schematic representation of the novel Biotin-Thiol-Assay (BTA), an experimental approach for the identification of sulfhydrated proteins in cell extracts. Highly reactive cysteine residues or sulfhydrated cysteine residues in proteins under native conditions were alkylated with low concentrations of maleimide-PEG2-biotin (NM-Biotin). Subsequent avidin column purification and elution with DTT, which cleaved the disulfide bonds, leaving the biotin tag bound to the column, produced an eluate that was further analyzed either by western blotting or coupled with LC-MS/MS. (**I**) Identification via the BTA of sulfhydrated GAPDH in MIN6 cell extracts from Tg-treated cells in the presence or absence of ATF4. (**J**) Identification of sulfhydrated GAPDH in MIN6 cells overexpressing ATF4. (**K**) Determination of the effect of PAG on sulfhydrated GAPDH levels in MIN6 cells overexpressing ATF4 in the presence or absence of PAG. ER: endoplasmic reticulum; PAG: propargylglycine.

The following source data and figure supplements are available for figure 2:

**Source data 1.** Sulfhydrated proteins from MIN6 cells treated with Tg for 18 hr.
**Figure supplement 1.** Expression of genes in MIN6 cells overexpressing the transcription factor ATF4.
**Figure supplement 2.** Schematic representation of the predicted proteins in the eluate of the BTA approach as a consequence of increasing concentrations of biotin conjugated maleimide (NM-biotin, red).

*Figure 2 continued on next page*

*Figure 2 continued*

**Figure supplement 3.** Increasing concentrations of NM-biotin in the BTA of cell extracts isolated from Tg treated for 18 hr MIN6 cells, inhibit the elution of sulfhydrated proteins.

**Figure supplement 4.** $H_2S$ covalently modifies proteins via sulfhydration of cysteine residues.

**Figure supplement 5.** Assessment of the specificity of BTA to identify reactive -S-SH groups of proteins via the use of recombinant GAPDH.

**Figure supplement 6.** Assessment of GAPDH sulfhydration by the red maleimide assay.

**Figure supplement 7.** BTA identifies sulfhydration of GAPDH at the catalytic cysteine, Cys[150].

**Figure supplement 8.** The BTA assay shows that Tg-induced ER stress in MIN6 cells promotes global protein sulfhydration.

**Figure supplement 9.** LC-MS/MS spectrum of $H_2S$-modified peptides purified by the BTA technique from MIN6 cells treated with Tg for 18 hr.

**Figure supplement 10.** Western blot analysis of eluates from the BTA of sulfhydrated GAPDH and PHGDH in MIN6 cells treated with Tg in a time-dependent manner. a peak of protein sulfhydration occurred at 12 h of Tg treatment.

**Figure supplement 11.** Western blot analysis for CTH protein levels from MIN6 cells infected with either adenovirus-mediated expression of GFP as control, or shRNA against the CTH mRNA in a dose.

synthesis with a parallel loss of induction of CTH protein levels (*Figure 2B*). In contrast, in the absence of stress, ATF4 overexpression increased CTH and Ero1α levels and $H_2S$ synthesis, in agreement with increased GAPDH activity (*Figure 2C–E*, *Figure 2—figure supplement 1*). The levels of GSH and the activity of the glutamate/cystine exchanger also increased with ATF4 overexpression (*Figure 2F, G*). These data support the hypothesis that ATF4 is a master regulator of protein sulfhydration in pancreatic β cells during ER stress.

To profile genome-wide changes in protein sulfhydration by ER stress and the integrated stress response, we exploited the different reactivity of cysteine persulfides (Cys-S-SH) and thiols (Cys-SH) to alkylating agents (*Nishida et al., 2012*; *Pan and Carroll, 2013*; *Paul and Snyder, 2012*), to develop a new thiol reactivity-based approach called BTA (Biotin Thiol Assay, (*Figure 2H* and *Figure 2—figure supplement 2*). We employed the following steps (*Figure 2H*): (1) labeling of reactive Cys-SH or Cys-S-SH groups by a biotin-conjugated maleimide (maleimide-PEG2-biotin, NM-biotin) (*Weerapana et al., 2010*), (2) binding of the biotin-labeled proteins on an avidin column, (3) elution of the retained proteins that contain a persulfide bridge using DTT, and (4) analysis of the eluted proteins.

The BTA approach was validated by testing the hypothesis that a lower concentration of the thiol-alkylating reagent (NM-biotin) will result in a selective labeling of highly reactive cysteine SH groups, and that this selectivity will be lost as the concentration of NM-biotin increases (*Weerapana et al., 2010*). We tested this hypothesis in extracts isolated from Tg-treated MIN6 cells in the presence of increasing concentrations of NM-biotin (0.05–1 mM). The eluate containing the sulfhydrated proteins was analyzed with SDS–PAGE and the proteins were visualized by Coomassie blue staining. Indeed, increasing the concentration of NM-biotin beyond 0.5 mM decreased the levels of eluted proteins (*Figure 2—figure supplement 3*), consistent with cysteine residues having free -SH groups becoming alkylated at high concentrations of NM-biotin. No protein was eluted from the column with the addition of DTT at high concentrations of NM-biotin (*Figure 2—figure supplement 3*), confirming that the biotin was attached to proteins via a disulfide bond (*Figure 2—figure supplement 4A*). This selectively labeling behavior not only relies on the probe concentration, but also is dependent on the protein structure conformations. We found that no protein was detected in the eluates if the BTA was performed on the denatured cell extracts (data not shown).

When Tg-treated MIN6 lysates were pretreated with DTT (to decrease the level of sulfhydrated proteins), the signal was significantly reduced compared to untreated lysates (*Figure 2—figure*

*supplement 4B*), confirming that DTT reduces an intermolecular disulfide bond of cysteine persulfides labeled with biotin. These data show that sulfhydrated cysteine residues are the primary targets of NM-biotin at low concentrations, thus making the BTA a unique tool to identify the sulfhydrated proteome (*Figure 2—figure supplement 2*).

We next assessed the BTA assay using recombinant GAPDH, which contains six cysteine residues including the redox-regulated $Cys^{150}$. GAPDH was incubated with either $H_2O_2$ or NaHS. The presence of sulfhydrated GAPDH was evaluated by the BTA method following Western blot analysis. We found that only the NaHS-treated GAPDH was eluted as an $H_2S$-modified target, indicating that the assay distinguishes sulfhydration or free SH groups from other forms of cysteine modifications (*Figure 2—figure supplement 5B*). This was independently confirmed with the red malemide assay (*Figure 2—figure supplement 6*), which discriminates between free -SH groups and sulfhydrated-SH (*Sen et al., 2012*). We next tested if the BTA can identify modification of a specific cysteine on GAPDH by using recombinant wild-type and $Cys^{150}$Ser GAPDH mutant, which were incubated with $H_2O_2$ or NaHS. We found that NaHS treatment induced sulfhydration in the wild-type GAPDH (*Figure 2—figure supplement 7A*), a result that was also confirmed by high-resolution quadruple MS analysis (*Figure 2—figure supplement 7B–C*), and that this modification was absent in the $Cys^{150}$Ser mutant. Finally, a proteome-wide view of sulfhydration was obtained from Tg-treated MIN6 cell extracts subjected to BTA and further analyzed by LC-MS/MS (*Figure 2—figure supplement 8*). We identified 150 proteins, including several known targets for sulfhydration (*Figure 2—figure supplement 8D*) (*Mustafa et al., 2009*). Similar results were obtained from analysis of mouse liver (data not shown), a tissue known to exhibit high levels of $H_2S$ synthesis (*Kabil et al., 2011*). Taken together, this shows that BTA discriminates between protein sulfhydration and other oxidative modifications.

Next, the BTA methodology identified the sulfhydrated proteome downstream of the transcription factor ATF4 during ER stress (*Figure 2A*). MIN6 cells treated with Tg increased GAPDH levels in the DTT eluate, that was abolished by knocking down ATF4 (*Figure 2I*). In the absence of stress, ATF4 knockdown resulted in an increase in sulfhydrated GAPDH in MIN6 cells (*Figure 2I*). Because ATF4-deficient cells have decreased levels of sulfur-containing amino acids (*Harding et al., 2003*), and sulfur amino acid restriction is linked to an increase in the transulfuration pathway (*Hine et al., 2015*), it is possible that the increase in GAPDH sulfhydration in ATF4-depleted cells is due to activation of CBS (cystathionine-β-synthase) (*Niu et al., 2015*), the second cytosolic $H_2S$-producing enzyme. Moreover, we found that in the absence of stress ATF4 overexpression induced GAPDH sulfhydration that is dependent on the CTH activity (*Figure 2J–K*). The use of PAG decreased GAPDH sulfhydration (*Figure 2K*). We conclude that sulfhydration of proteins during ER stress is part of the ISR and is controlled by the transcription factor, ATF4.

The BTA requires determining the concentration of NM-Biotin for selectively labeling proteins with reactive, sulfhydrated cysteines rather than the relatively high abundant and less reactive, unmodified (with free SH groups) cysteine residues. However, this labeling step has some limitations under certain circumstances. For example, if free Cys-SH groups are biotinylated on the same protein containing one or more Cys-S-SH groups (*Figure 2H* and *Figure 2—figure supplement 2*), then the protein will not be eluted with DTT and will not be identified as a target for sulfhydration. In addition, if a protein contains sulfhydrated cysteines with low reactivity for the probe, this protein will not be captured and identified as an $H_2S$-modified target. Due to those limitations and in order to extend the capability of the BTA approach, we introduced a proteolytic digestion step before applying the avidin column step. This added step provided not only the isolation of cysteine-containing peptides with persulfide bonds, but also increased the identification of sulfhydrated proteins. The eluted peptides were then sequenced and identified by LC-MS/MS analysis, thereby identifying the modified cysteines on proteins. By using the modified BTA technique, we have identified over ~ 1000 novel sulfhydrated cysteines in MIN6 cells treated with Tg, corresponding to about 820 proteins (*Figure 2—figure supplement 9* and *Figure 2—source data 1*), including GAPDH, wherein two cysteine-containing peptides were captured: $Cys^{150}$ and $Cys^{245}$. Remarkably, the $Cys^{150}$ active-site peptide was highly enriched as compared to the C-terminal of $Cys^{245}$, supporting prior mutagenesis studies that have shown $Cys^{150}$ as the primary $H_2S$-modified site on GAPDH, both in vitro and in vivo. One of novel targets including phosphoglycerate dehydrogenase (PHGDH) was also confirmed by Western blot analysis (*Figure 2—figure supplement 10*). CTH Knockdown mediated by shRNA decreased GAPDH and PHGDH sulfhydration levels, confirming that those proteins identified

by the BTA method are bona-fide targets for sulfhydration in vivo (*Figure 2—figure supplement 11*). Sulfhydrated peptides do not reveal a consensus protein abundance and sulfhydrome (*Figure 3—figure supplement 6,7* and *Figure 3—source data 2*), supporting that lower concentrations of the NM-biotin labeling reveal reactivity of cysteine residues rather than protein abundance.

In order to obtain genome-wide stress-induced changes in the sulfhydrome and individual cysteine residues within these proteins, we devised a modified BTA protocol (*Figure 3A*) by introducing a stable isotope-labeling step after the DTT elution step. This protocol uses (1) digestion of biotinylated cell extracts with trypsin before avidin capture, (2) labeling of free Cys-SH groups on peptides in the column eluent by mass-difference cysteine-alkylating reagents with either NEM-H$_5$ (Light) or NEM-D$_5$ (Heavy), and (3) quantification by LC-MS/MS analysis of H/L ratios of the individual pair-labeled cysteines in the identified peptides based on a mass-difference labeling. In addition to quantifying changes in protein sulfhydration, this modified BTA approach allows the detection of additional proteins (*Figure 2—figure supplement 11* and *Figure 2—source data 1*). We used as an experimental system the ATF4-expressing MIN6 cells to profile quantitatively the sulfhydrated proteome. As shown in *Figure 2A*, ATF4-mediated signaling triggers the cellular response, which leads to increased protein sulfhydration. Using the experimental plan in *Figure 3A*, we identified over 834 cysteine-containing peptides (*Figure 3B* and *Figure 3—source data 1*). Of these peptides, 771 exhibited pair-labeling with an overall average H/L ratio 2.6, and 348 peptides (45%) displayed high ratios (H/L>2). These findings confirmed that ATF4 drives global changes in protein sulfhydration in MIN6 cells.

Sulfhydrated cysteine residues in proteins may contribute to their biological activities, especially when these modified cysteine residues reside within functional domains. We thus queried the Uniprot database to retrieve functional annotations for the aforementioned 827 peptides. The analysis revealed that 28% of the peptides were localized to protein regions whose structural and functional properties are known (*Figure 3C*, *Figure 3—figure supplement 1*). An additional 18% were found within functional regions of proteins with cysteine residues of unknown significance. Our finding of sulfhydration of specific cysteine residues within functional domains of proteins suggests that the cysteine modification influences the activity of proteins. In contrast, 4.2% of the peptides in question contained cysteine residues in experimentally proven active sites of enzymes or cysteine residues involved in disulfide bond formation (*Figure 3C*). This percentage of active cysteine-containing peptides from the BTA assay is significantly larger than the 0.2% of all cysteines in the Uniprot database that have been assigned to experimentally characterized active cysteines (*Weerapana et al., 2010*). Finally, 2.4% of the annotated peptides were found in known nucleotide binding domains of proteins, suggesting the potential for regulation of gene expression by H$_2$S cysteine-modified proteins. None of these proteins have been reported as targets of sulfhydration. Although not found among the 827 peptides in this study, the NF-κB RelA transcription factor has been shown to be sulfhydrated in the DNA-binding domain, resulting in increased DNA-binding activity (*Sen et al., 2012*). We also found by querying the redox modification databases (RedoxDB and GPS-SNO) that 11% of the 834 peptides from the RedoxDB and 36% from the GPS-SNO corresponded to cysteine residues that are known to be modified by nitrosylation or glutathionylation (*Figure 3B*, *Figure 3—figure supplement 2*). Taken together, these data indicate that protein sulfhydration not only influences the catalytic activity of enzymes like GAPDH, but also can potentially regulate a broad range of biological processes.

We noted a wide range of H/L ratios in the identified peptides, reflecting a large difference in the reactivity of individual cysteines to sulfhydration in MIN6 cells that overexpress ATF4 (*Figure 3B–D*). In addition, labeled cysteines on the same protein exhibited remarkably different ratios (*Figure 3—figure supplement 3*). For example, the protein, PHGDH, was labeled on two cysteines, Cys[281] had an H/L ratio of 5.8, whereas Cys[254] displayed a ratio of 17. To identify a sulfhydration motif in proteins, all cysteine-containing peptides (*Figure 3B*) were analysed by the pLogo motif analyzer program. This analysis did not reveal conserved residues surrounding the modified cysteine site (*Figure 3—figure supplement 4A*), a result consistent with other enzyme-independent oxidative modifications of cysteines (*Weerapana et al., 2010*) (*Marino and Gladyshev, 2011*). However, when we selected only the peptides with high H/L ratios (>2), and searched for a sequence motif at the cysteine modification sites (*Figure 3—figure supplement 4B*), an Arg residue was significantly enriched at the +1 position of the modified cysteine (*Figure 3—figure supplement 4B*). Additionally, structure motif and surface accessibility analysis revealed that the modified cysteine is highly

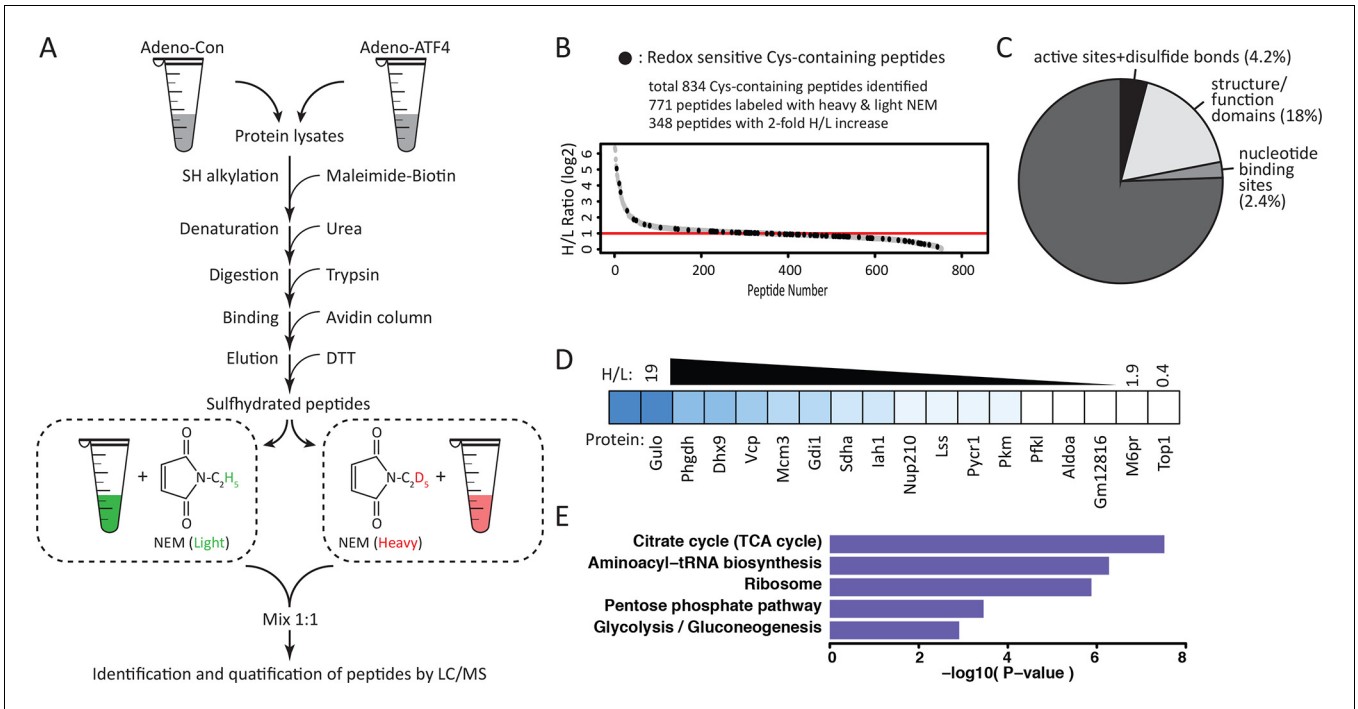

**Figure 3.** Quantitative and pathway analysis of sulfhydrated peptides in MIN6 cells overexpressing the transcription factor ATF4. (**A**) Schematic representation of the BTA experimental approach combined with alkylation of free SH groups by either the stable isotope-labeled ($D_5$, heavy) or normal ($H_5$, light) maleimide. The relative levels of $H_5$ and $D_5$ labeled peptides were quantified by the LC-MS/MS method. (**B**) Distribution of peptides containing sulfhydrated cysteine residues relative to their H/L ratios as determined by the BTA analysis of cell extracts isolated from MIN6 cells overexpressing ATF4 (indicated in **A**). Values of H/L ratios are plotted against the number of identified peptides. The red line marks the H/L ratio >2, consisting of cysteine-containing peptides in proteins that exhibited higher reactivity with $H_2S$ under ATF4 overexpression. The black dots show redox sensitive cysteine peptides, which are common between the ones found in the RedoxDB database, and by the BTA assay. (**C**) Pie chart illustrating the percentage of cysteine-containing peptides (from **A**) that belong to known functional domains of proteins in the Uniprot database. (**D**) Heat map of H/L values obtained from experimental data in (**A**), illustrating the profound differences in the reactivity with $H_2S$ of cysteine residues in different proteins. (**E**) Gene ontology biological pathways for peptides with H/L ratio >2. H/L values were obtained from the experimental data in (**A**). BTA: biotin thiol assay.

The following source data and figure supplements are available for figure 3:

**Source data 1** Sulfhydrated proteins from ATF4 overexpressed MIN6 cells.

**Source data 2.** Relative abundance of proteins in MIN6 cells treated with Tg for 18 hr.

**Figure supplement 1.** Sulfhydrated cysteine-containing peptides are enriched in functional residues.

**Figure supplement 2.** Sulfhydrated proteins are potential targets for nitrosylation.

**Figure supplement 3.** Quantitative profiling of proteins containing cysteines with different reactivity to $H_2S$ from ATF4 overexpressing MIN6 cells.

**Figure supplement 4.** (A,B) Sulfhydrated peptides do not reveal a consensus sequence motif, (C,D) but the modified cysteine residue is significantly accessible and preferentially positioned at the N-terminal of alpha helix.

**Figure supplement 5.** Gene ontology biological pathways enriched among all pair-labeled peptides in ATF4 overexpressed MIN6 cells.

**Figure supplement 6.** MS analysis of the full proteome from MIN6 cells treated with Tg for 18 hr.

**Figure supplement 7.** Protein sulfhydration does not correlate with their protein abundance.

accessible and positioned at the N-terminal alpha-helices (*Figure 3—figure supplement 4C–D*). This

is consistent with previous reports suggesting that a reactive cysteine thiolate anion is stabilized by interaction with alpha-helix dipoles (*Kortemme and Creighton, 1995*; *Weerapana et al., 2010*).

Bioinformatics clustering with two pathway annotation programs DAVID and Ingenuity Pathway Analysis (IPA) revealed an enrichment of sulfhydrated proteins in glycolysis and mitochondrial oxidative metabolism (*Figure 3E*, *Figure 3—figure supplement 5*). Data from this analysis are in agreement with increased activity of the glycolytic enzyme, GAPDH, by sulfhydration, and prompted the question as to whether or not glycolytic flux is regulated by the $H_2S$-dependent modification of enzymes involved in intermediary metabolism. Also in agreement with this hypothesis, we found that ATF4-overexpressing MIN6 cells had higher glycolytic rates as evaluated with a Seahorse analyzer (*Figure 4—figure supplement 1*). Furthermore, the activity of the rate-limiting glycolytic enzyme, pyruvate kinase 2 (PKM2), also was increased in ATF4 overexpressing MIN6 cells, in a manner dependent on CTH activity (*Figure 4—figure supplement 2*). We therefore returned to the induction of ER stress by Tg-treatment of MIN6 cells and evaluated glycolytic flux rates in the absence or presence of PAG. The advantage of using PAG instead of genetic manipulation such as gene knockdown is that the inhibitor could be added at the same time as labeled glucose, thus allowing assessment of the inhibitor's effect on glycolytic flux rates.

We directly measured changes in metabolic flux by utilizing stable isotope label incorporation and mass isotopomer analyses. MIN6 cells were treated with Tg for 18 hr; the growth media was changed to (D-glucose-$^{13}C_6$) media in the presence or absence of PAG during the last 3.5 hr of treatment. Tg-treatment significantly augmented glycolytic flux as determined by the increase in the glycolytic intermediate, 3 phosphoglyceric acid (3PG), as well as lactate and alanine (*Figure 4A–C*, *Supplementary file 1*). The flux was consistent with an increase in the relative concentrations of 3PG and alanine (*Figure 4B*, *Supplementary file 1*). However, lactate levels were decreased significantly despite the increased flux. A decrease in cellular lactate levels supports the idea that there is high consumption of pyruvate by the mitochondria to generate oxaloacetate (OAA). In contrast to the increased glycolytic rates, flux to tricarboxylic acid cycle (TCA) intermediates was significantly reduced by Tg-treatment as evidenced by the low $^{13}C$ label incorporation of both acetyl-CoA and OAA moieties of citrate, fumarate, and malate (*Figure 4A–B*, *Supplementary file 1*). When the cells were exposed to PAG along with Tg-treatment, the increase in glycolytic flux was prevented, as shown by the decrease in $^{13}C$-labeling of 3PG, lactate, and alanine. In contrast, TCA cycle flux was restored, as exemplified by the increase in $^{13}C$-labeling of both OAA and acetyl-CoA moieties of citrate, succinate, fumarate, and malate, and a decrease in their concentrations, suggesting utilization. $^{13}C$-labeling of aspartate and glutamine also increased significantly, indicating increased cataplerosis of TCA cycle intermediates. Moreover, we determined the activity of PKM2 in MIN6 cells treated with Tg in the presence or absence of PAG. Tg-treatment increased PKM2 activity, but PAG addition inhibited the increase, without affecting PKM2 protein levels (*Figure 4—figure supplement 3*). These data suggest that ER stress, via $H_2S$-mediated signaling, promotes glycolysis and decreases mitochondrial oxidative metabolism.

## Discussion

In summary, sulfhydration of specific cysteines in proteins is a key function of $H_2S$ (*Kabil and Banerjee, 2010*; *Paul and Snyder, 2012*; *Szabo et al., 2013*). Thus, the development of tools that can quantitatively measure genome-wide protein sulfhydration in physiological or pathological conditions is of central importance. However, a significant challenge in studies of the biological significance of protein sulfhydration is the lack of an approach to selectively detect sulfhydrated cysteines from other modifications (disulfide bonds, glutathionylated thiols and sulfenic acids) in complex biological samples. In this study, we introduced the BTA approach that allowed the quantitative assessment of changes in the sulfhydration of specific cysteines in the proteome and in individual proteins. BTA is superior to other reported methodologies that aimed to profile cysteine modifications, such as the most commonly used, a modified biotin switch technique (BST). BST was originally designed to study protein nitrosylation and postulated to differentiate free thiols and persulfides (*Mustafa et al., 2009*). A key advantage of BTA over the existing methodologies is that the experimental approach has steps to avoid false-positive and negative results, as target proteins for sulfhydration. BST is commonly generating such false targets for cysteine modifications (*Forrester et al., 2009*; *Sen et al., 2012*).

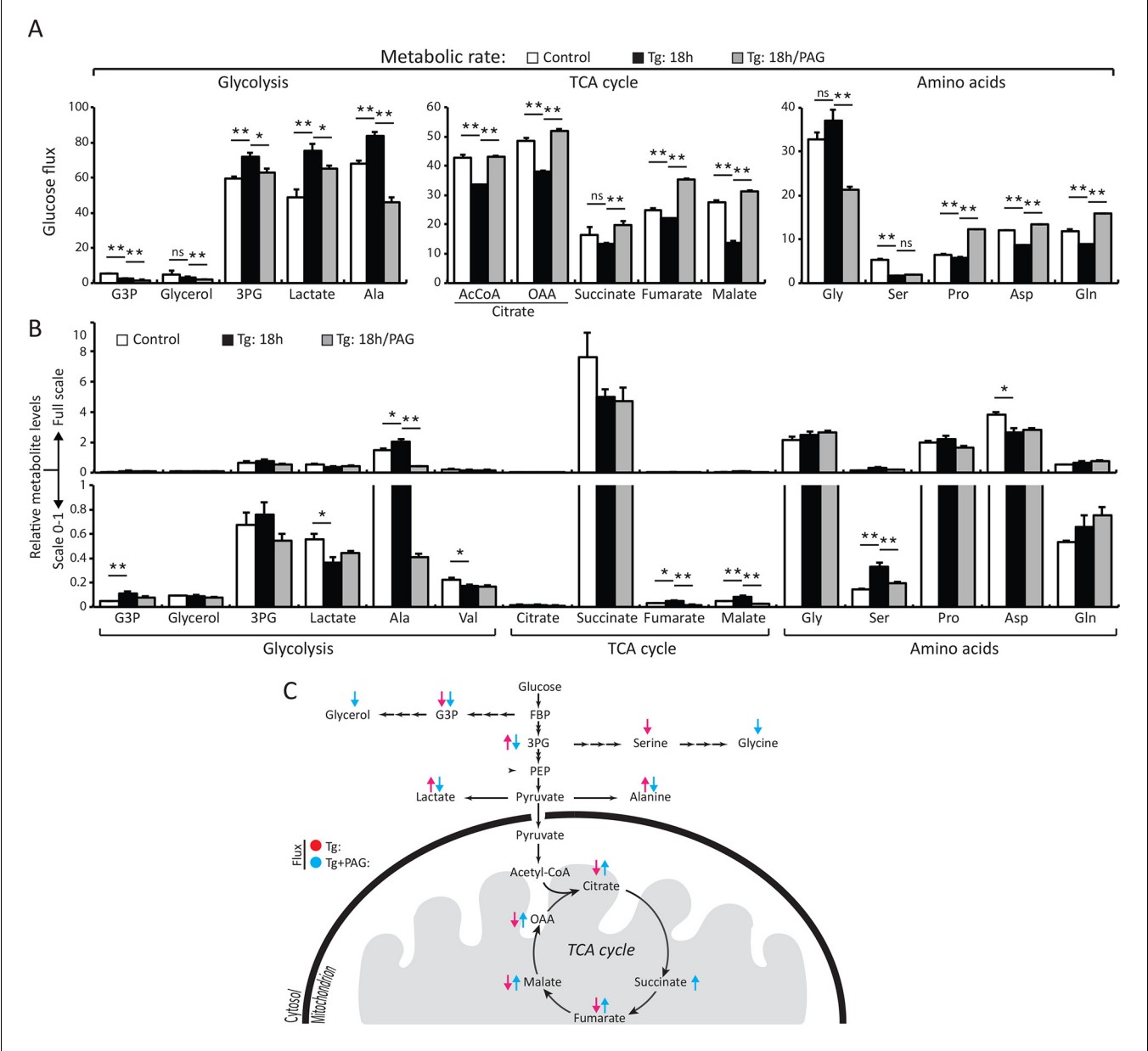

**Figure 4.** $H_2S$ synthesis during ER stress modulates metabolism in MIN6 cells. (**A**) Measurement of $^{13}C$-glucose flux in metabolites, expressed as the molar percent enrichment [(ratio of labeled/sum (labeled + unlabeled) x 100%)], in MIN6 cells treated with Tg for 18 hr or after addition of PAG for the last 3.5 hr of Tg treatment. [U-$^{13}C$]-glucose replaced glucose in the media for the last 3.5 hr of treatments. (**B**) Evaluation of the concentration of metabolites and amino acids in the same samples described in (**A**). All quantifications are presented as mean ± S.E.M. of technical duplicates and are represented four independent experiments. (**C**) Schematic representation of the major findings on metabolic flux from glucose during chronic ER stress. Chronic ER stress increased glycolytic flux and decreased forward TCA flux. Inhibition of CTH by PAG reversed the observed changes in glucose flux during ER stress. CTH: γ-cystathionase; ER: endoplasmic reticulum; TCA: tricarboxylic acid cycle.

The following figure supplements are available for figure 4:

**Figure supplement 1.** ATF4 -overexpressing MIN6 cells exhibit significantly high glycolytic rates.

**Figure supplement 2.** Pharmacological inhibition of CTH activity represses PKM2 activation in ATF4 overexpressed MIN6 cells.

**Figure supplement 3.** PKM2 activation dependents on CTH activity during ER stress in MIN6 cells.

Using mutiple validations, our data support the specificity and reliability of the BTA assay for analysis of protein sulfhydration both in vitro and in vivo. With this approach, we found that ATF4 is the master regulator of protein sulfhydration in pancreatic β cells during ER stress, by means of its function as a transcription factor. A large number of protein targets have been discovered to undergo sulfhydration in β cells by the BTA approach. Almost 1000 sulfhydrated cysteine-containing peptides were present in the cells under the chronic ER stress condition of treatment with Tg for 18 hr. Combined with the isotopic-labeling strategy, almost 820 peptides on more than 500 proteins were quantified in the cells overexpressing ATF4. These data show the potential of the BTA method for further systematic studies of biological events. To our knowledge, the current dataset encompasses most known sulfhydrated cysteine residues in proteins in any organism. Our bioinformatics analyses revealed sulfhydrated cysteine residues located on a variety of structure–function domains, suggesting the possibility of regulatory mechanism(s) mediated by protein sulfhydration. Structure and sequence analysis revealed consensus motifs that favor sulfhydration; an arginine residue and alpha-helix dipoles are both contributing to stabilize sulfhydrated cysteine thiolates in the local environment.

Pathway analyses showed that $H_2S$-mediated sulfhydration of cysteine residues is the part of the ISR with the highest enrichment in proteins involved in energy metabolism. The metabolic flux revealed that $H_2S$ promotes aerobic glycolysis associated with decreased oxidative phosphorylation in the mitochondria during ER stress in β cells. The TCA cycle revolves by the action of the respiratory chain that requires oxygen to operate. In response to ER stress, mitochondrial function and cellular respiration are down-regulated to limit oxygen demand and to sustain mitochondria. When ATP production from the TCA cycle becomes limited and glycolytic flux increases, there is a risk of accumulation of lactate from pyruvate. One way to escape accumulation of lactate is the mitochondrial conversion of pyruvate to oxalacetic acid (OAA) by pyruvate carboxylase. This latter enzyme was found to be sulfhydrated, consistent with the notion that sulfhydration is linked to metabolic reprogramming toward glycolysis.

The switch of energy production from mitochondria to glycolysis is known as a signature of hypoxic conditions. This metabolic switch has also been observed in many cancer cells characterized as the Warburg effect, which contributes to tumor growth. The Warburg effect provides advantages to cancer cell survival via the rapid ATP production through glycolysis, as well as the increasd conversion of glucose into anabolic biomolecules (amino acid, nucleic acid, and lipid biosynthesis) and reducing power (NADPH) for regeneration of antioxidants. This metabolic response of tumor cells contributes to tumor growth and metastasis (*Vander Heiden et al., 2009*). By analogy, the aerobic glycolysis trigged by increased $H_2S$ production could give β cells the capability to acquire ATP and nutrients to adapt their cellular metabolism toward maintaining ATP levels in the ER (*Vishnu et al., 2014*), increasing synthesis of glycerolphospholipids, glycoproteins and protein (*Krokowski et al., 2013*), all important components of the ISR. Similar to hypoxic conditions, a phenotype associated with most tumors, the decreased mitochondria function in β cells during ER stress, can also be viewed as an adaptive response by limiting mitochondria ROS and mitochondria-mediated apoptosis. We therefore view that the $H_2S$-mediated increase in glycolysis is an adaptive mechanism for survival of β cells to chronic ER stress, along with the improved ER function and insulin production and folding, both critical factors controlling hyperglycemia in diabetes. Future work should determine which are the key proteins targeted by $H_2S$ and thus contributing to metabolic reprogramming of β cells, and if and how insulin synthesis and secretion is affected by sulfhydration of these proteins during ER stress.

Abnormal $H_2S$ metabolism has been reported to occur in various diseases, mostly through the deregulation of gene expression encoding for $H_2S$-generating enzymes (*Wallace and Wang, 2015*). An increase of their levels by stimulants is expected to have similar effects on sulfhydration of proteins like the ATF4-induced CTH under conditions of ER stress. It is the levels of $H_2S$ under oxidative conditions that influence cellular functions. In the present study, ER stress in β cells induced elevated CTH levels, whereas CBS was unaffected. The deregulated oxidative modification at cysteine residues by $H_2S$ may be a major contributing factor to disease development. In this case, it would provide a rationale for the design of therapeutic agents that would modulate the activity of the involved enzymes.

## Materials and methods

### Mouse islets RNA isolation

Experimental protocols were approved by the Case Western Reserve University Institutional Animal Care and Use Committee. C57BL/6J and C57BL/6-*Ins2*[+/Akita] mice were used for experiments. Mice from the Jackson Laboratory were bred at the animal facilities at Case Western Reserve University and were fed standard lab chow (LabDiet). Mice were housed under 12:12 hr light/dark cycle with free access to food and water at 23°C. Mouse pancreatic islets were isolated as described before (*Krokowski et al., 2013*). Islets from 6 weeks old male C57BL/6-*Ins2*[+/Akita] (n=6) and age and sex matched wild type (n=6) were cultured for 2 hr in RPMI 1640 media supplemented with 10% FBS and 5 mM glucose before RNA isolation. For Tg treatment (1 µM), islets from wild-type mice (n=6) were combined and cultured in RPMI 1640 medium supplemented with 10% FBS in atmosphere of 5% $CO_2$ at 37°C for 24 hr. From each group 150–200 islets were manually picked and used for RNA isolation. Islets were treated with QIAshredder (Qiagen GmbH, D-40724 Hiden, Germany), and RNA was purified using the RNeasy Plus Micro kit (Qiagen GmbH, D-40724 Hiden, Germany).

### Human islets RNA isolation

Institutional review board approval for research use of isolated human islets was obtained from the University of Michigan. Human islets were isolated from previously healthy, nondiabetic organ donors by the University of Chicago Transplant Center. The islets were divided into two groups, incubated in CMRL medium containing either 5.5 mM glucose with or without Tg (1 µM), for 24 hr. The islets were frozen at -80°C before analysis. RNA was isolated as described above from 200 islets/treatment.

### RT-qPCR analysis of mRNAs for MIN6 cells

RNA was isolated from mouse MIN6 cells using TRIzol (Invitrogen). cDNA was synthesized from total RNA isolated from islets or MIN6 cells using the SuperScript III First-Strand Synthesis Super Mix (Invitrogen), and the abundance of cDNA isolated from each sample was quantified by qPCR using the VeriQuest SYBR Green qPCR Master Mix (Affymetrix) with the StepOnePlus Real-Time PCR System (Applied Biosystems).

### Cell culture and viral particles

MIN6 cells were cultured in high glucose DMEM supplemented with 10% FBS, 2 mM l-glutamine, 1 mM sodium pyruvate, 55 µM β-mercaptoethanol, 100 units/ml penicillin, and 100 mg/ml streptomycin at 37°C in atmosphere of 5% $CO_2$. β-mercaptoethanol was removed from the media 12 hr before experimentation. Rat INS1 cells were cultured in RPMI 1640 supplemented with 11 mM glucose, 10% heat inactive FBS, 2 mM l-glutamine, 1 mM sodium pyruvate, 100 units/ml penicillin, and 100 µg/ml streptomycin at 37°C in atmosphere of 5% $CO_2$. Tg (Sigma-Aldrich) was used at 400 nM and the CTH inhibitor - DL-propargylglycine (PAG, Sigma Aldrich) at 3 mM. Lentiviral particles were prepared in HEK293T as described before (*Saikia et al., 2014*). Lentiviral vector expressing shRNA against *ATF4* were obtained from Sigma-Aldrich (TRCN0000301646). Adenovirus mediated shRNA against mouse CTH (shRNA sequence: CCGGCCATTACGATTACCCATCTTTCTCGAGAAAGATGG-GTAATCGTAATGGTTTTTG) was purchased from Vector Biolabs. MIN6 cells were infected in the presence of 10 µg/ml polybrene and selection with 2.5 µg/ml puromycin (Life Technologies) was conducted 24 hr post-infection for 5 days. Adenovirus particles for expression of β galactosidase (β-Gal), GFP or mouse ATF4 protein were prepared in HEK293 cells and were used for infection as described before (*Guan et al., 2014*).

### Bacterial expression of wild type and Cys[150]Ser human recombinant GAPDH

Human GST-tagged wild type or C[150]S GAPDH mutant (*Hara et al., 2005*) was expressed in the *E. coli* BL21 strain. Protein expression was induced by addition of IPTG (100 µM). When bacterial cultures reached $OD_{600}$ of 0.8 at 37°C, IPTG was added for 4 hr incubation before lysis in a buffer containing 50 mM Tris–HCl (pH 7.5) and 1 mM EDTA. Lysates were centrifuged and applied on a buffer-equilibrated GST-sepharose affinity spin column (Pierce). After extensive washes to remove unbound

protein, recombinant GAPDH was released by digestion with thrombin protease (Sigma). The protein purity was determined on SDS–PAGE gels stained by Coomassie blue.

## GAPDH activity assay

The specific activity of GAPDH was determined as described before (*Hara et al., 2005*). Recombinant protein (50 nM) was used for the in vitro activity assays. To test the GAPDH activity in cell lysates, MIN6 cells were harvested in RIPA buffer (150 mM NaCl, 1 mM EDTA, 1% Triton X-100, 0.5% deoxycholic acid, 50 mM Tris–HCl, pH 7.5), sonicated on ice and centrifuged at 4°C. One to twenty microrams of protein lysate was used for the activity assays. The reaction mixture contained 100 mM Tris–HCl (pH 7.5), 5 mM $MgCl_2$, 3 mM 3-phosphoglycerate, 5 units/ml of *Saccharomyces cerevisiae* 3-phosphoglycerate kinase (PGK, Sigma-Aldrich), 2 mM ATP and 0.25 mM NADH. Reactions were conducted in 200 μl volume at 25°C and monitored spectrophotometrically at 340 nm for 3 min using a M3 microplate reader (Molecular Devices).

## PKM activity assay

PKM activity was tested in the cell extracts as described (*Anastasiou et al., 2011*). MIN6 cells were collected by trypsinization and the cell pellets were washed twice with cold PBS. Cells were resuspended in RIPA buffer, sonicated on ice, and subjected to a quick centrifugation at 4°C. Reaction mixtures contained 50 mM Tris–HCl (pH 7.5), 100 mM KCl, 5 mM $MgCl_2$, 0.5 mM ADP, 0.2 mM NADH, 8 units of lactate dehydrogenase (Sigma-Aldrich), and 1–10 μg of cell lysate. The enzymatic reaction was initiated by the addition of PEP (phosphoenolpyruvic acid, 0.5 mM) as the substrate. The oxidation of NADH was monitored at 340 nm for 3 min using a M3 microplate reader (Molecular Devices).

## Activity of cystine/glutamate exchanger Slc7a11

The activity of the amino acid transporter was tested as described before (*Krokowski et al., 2013*). Uptake of Glu was tested in the absence of $Na^+$ ions (EBSS solution, NaCl replaced with choline chloride), with 100 μM Glu and 4 μCi/ml of $^3$H-Glu (Parkin Elmer) for 3 min at 37°C. MIN6 cells were washed twice with cold PBS then amino acids were extracted with ethanol and radioactivity was counted. The specific activity was normalized to protein content that was determined by the Lowery assay.

## Intracellular glutathione content and the ratio of oxidized and reduced glutathione

MIN6 cells ($6x10^4$ cell per well) were seeded into 96-well plates. After 48 hr the growth media was removed, the total glutathione contents and GSH/GSSG ratios were determined with the GSH/GSSG-Glo Tm assay from Promega.

## ROS measurements

Total intracellular levels of ROS were quantified using dichlorofluorescein diacetate (CM-$H_2$DCFDA; 10 μM). MIN6 cells were seeded into 96-well plates. After 48 hr, cells were washed with warm PBS and incubated with the dye in phenol-red free DMEM without FBS. After 1 hr, the cells were washed with PBS to remove the dye and placed in phenol-red free DMEM. CM-$H_2$DCFDA fluorescence was measured at excitation/emission wavelengths of 495/517 nm. Cells not exposed to the probe were used to test the background fluorescence. After subtraction of background fluorescence results were normalized to protein content determined by the BCA assay.

## $H_2$S-production assays

Frozen cell pellets were lysed in 100 mM HEPES (pH 7.4), to obtain a lysate concentration of 100 mg/ml. $H_2$S production was measured as described previously (*Kabil et al., 2011*). Briefly, reactions containing cell lysate (200 μl), 10 mM cysteine and 100 mM HEPES (pH 7.4) were prepared in 20-ml polypropylene syringes in a total reaction volume of 400 μl. Reactions were started with the addition of cysteine. Syringes were sealed and the headspace was flushed with nitrogen five times by using a three-way stopcock, and left in nitrogen in a final total volume (aqueous + gas) of 20 ml. Syringes were placed at 37°C in a shaker incubator (75 rpm) for 20 min. Control reactions without cell lysates

were prepared in parallel. Aliquots of 0.2 ml from the gas phase were collected through a septum attached to the stopcock, and injected in an HP 6890 gas chromatograph (GC) (Hewlett Packard) equipped with a DB-1 column (30 m×0.53 mm×1.0 µm). Flow rate of the carrier gas (helium) was 1 ml/min, and the temperature gradient ranged from 30°C to 110°C over a 20-min period. $H_2S$ was detected by a 355 sulfur chemiluminescence detector (Agilent) attached to the GC. $H_2S$ standard gas (Cryogenic Gases, Detroit, MI) with a stock concentration of 40 ppm (1.785 µM) in nitrogen was used to generate a standard curve. The amount of $H_2S$ in the injected volume was calculated from the peak areas by using the calibration coefficient obtained from the standard curve. Ionized $H_2S$ concentration in the liquid phase was calculated for the pH of the reaction mixture (pH 7.4) by using a p$K$a value of 6.8 for ionization of $H_2S$. The resulting $H_2S$ concentration in the total reaction volume was then used to obtain the specific activity expressed as nmol $H_2S$ per mg protein per min.

## Biotin thiol assay

In order to detect and identify sulfhydrated proteins from MIN6 cells, cells were lysed with the RIPA buffer (150 mM NaCl, 1 mM EDTA, 0.5% Triton X-100, 0.5% deoxycholic acid, and 100 mM Tris–HCl (pH 7.5) containing protease and phosphatase inhibitor from Roche. Cells were sonicated on ice, lysates were clarified by centrifugation at 4°C and the protein concentrations were determined by the BCA assay (BioRad). Equal amount (4 mg) of proteins was incubated with 100 µM NM-biotin (Pierce) for 30 min with occasionally gentle mixing at room temperature and subsequently precipitated by cold acetone. After centrifugation, pellets of precipitated proteins were washed with 70% cold acetone twice, and then suspended in buffer (0.1% SDS, 150 mM NaCl, 1 mM EDTA and 0.5% Triton X-100, 50 mM Tris–HCl, pH 7.5) mixed with Streptavidin-agarose resin (Thermo Scientific) and kept rotating overnight at 4°C. The beads were washed five times with wash buffer 1 (0.5% Triton x-100, 150 mM NaCl, 50 mM Tris–HCl, pH 7.5) followed by five washes with wash buffer 2 (0.5% Triton X-100, 600 mM NaCl, 50 mM Tris–HCl, pH 7.5). Resin with bound proteins was incubated with 500 µl of the elution buffer with or without 20 mM DTT for 30 min at 25°C. Eluted proteins were concentrated to a final volume of 25–40 µl with utilization of Amicon Ultracel 10K (Millipore) and used for gel electrophoresis followed by either western blot or MS analysis.

## Red maleimide assay

The assay was previously described (*Sen et al., 2012*) and modified in order to adapt to our experimental needs. Purified recombinant GAPDH was next treated with either 50 µM NaCl as control, 50 µM NaHS or 50 µM $H_2O_2$ for 45 min at room temperature. After desalting through a spin column (Pierce), the samples were incubated with 1 µM red maleimide probe (Alexa Fluor 680 $C_2$ Maleimide, Molecular Probes) for 20 min at room temperature. After the incubation, these samples were treated with or without 10 mM β-mercaptoethanol and the reaction was stopped by the addition of 100 mM iodoacetamide. Samples were suspended in sample buffer for non-reducing gel electrophoresis. After electrophoretic separation, the gel was scanned with the Li-COR Odyssey system. The intensity of red fluorescence of GAPDH was quantified with the Odyssey system software. Subsequently proteins from the gel were transferred on a PVDF membrane and subjected to Western blot analysis for GAPDH.

## Purification of $H_2S$-modified cysteine-containing peptides from MIN6 cells

In order to identify $H_2S$-modified cysteine-containing peptides from cell lysates, proteins were extracted and biotinylated as described above. Biotinylated proteins were precipitated with ice cold acetone, resuspended in denaturation buffer (30 mM Tris–HCl (pH 7.5), 8 M urea and 1 mM $MgSO_4$) as described (*Morisse et al., 2014*), diluted with 10 volumes of the buffer (30 mM Tris–HCl (pH 7.5), 1 mM EDTA and 200 mM NaCl) and incubated with modified porcine trypsin (Promega) with occasionally mixing for 18 hr at 30°C. The ratio of the enzyme to substrate was 1:80 (w/w). After digestion, trypsin was inactivated by incubation at 95°C for 10 min then reactions were mixed with the streptavidin-agarose beads (500 µl) and incubated at 4°C for 18 hr following extensive washes in the presence of 0.1% SDS as described above. Peptides were eluted with 20 mM ammonium bicarbonate supplemented with 10 mM DTT after 25 min incubation at room temperature. DTT was removed with utilization of a C-18 column (Pierce). Peptides were eluted from the desalting column with

acetonitrile, dried under vacuum and suspended in buffer (30 mM Tris–HCl (pH 7.5), 1mM EDTA and 150 mM NaCl). Free -SH groups were alkylated by NEM (either deuterium or hydrogen containing) at final concentrations of 40 mM. The alkylated peptides were concentrated with a C-18 column (Pierce) for LC-MS/MS analysis.

## Liquid-chromatography-mass-spectrometry (LC-MS/MS) analysis

LC-MS/MS analysis was performed on an LTQ-Orbitrap Elite mass spectrometer (Thermo-Fisher) coupled to an Ultimate 3000 high-performance liquid chromatography system. Protein digests were loaded onto a 75 µm desalting column packed with 2 cm of Acclaim PepMap C18 reverse phase resin (Dionex). The peptides were then eluted onto a Dionex 15 cm x 75 µm id Acclaim Pepmap C18, 2µm, 100 Å reversed- phase capillary chromatography column using a gradient of 2–80% buffer B in buffer A (buffer A: 0.1% formic acid; buffer B: 5% water, 95% acetonitrile, 0.1% formic acid). The peptides were then eluted from the C18 column into the mass spectrometer at a flow rate of 300 nl/ min and the spray voltage was set to 1.9 kV. One full MS scan (FTMS) (300–2,000 MW) was followed by 20 data dependent scans (ITMS) of the $n$th most intense ions with dynamic exclusion enabled.

## Peptide identification

Peak lists were extracted from Xcaliber RAW files using Proteome Discoverer 1.4. These peak lists were searched Sequest HT and Mascot (2.3) search engines. The data was searched against the mouse reference sequence database which contains 77,807 entries using a precursor ion tolerance of 10 ppm and a fragment ion tolerance of 0.6 Da. These searches included differential modification of +125.047679 and +130.079062 on cysteine to account for NEM and $d_5$-NEM alkylation and +15.994915 Da to account for oxidation on methionine residues. Peptide identification was validated with the Percolator node on the basis of q-values which are estimated from target-decoy searches. The false discovery rate (FDR) for these searches was set to 1% at the peptide level. In addition, peptides were also required to be fully tryptic and have Xcorr scores > 1.5 (+1), 2.0 (+2), 2.25 (+3), and 2.5 (+4).

## Ratio quantification

Quantification of light/heavy ratios ($d_5$-NEM/NEM) was performed using two algorithms of Proteome Discoverer, the event detector and Precursor Ions Quantifier. The event detector applied a 2 ppm mass variability and 0.2 min chromatographic window for the generation of extracted ion chromatograms. The Peptide ratio was calculated from the summed extracted ion chromatograms of all isotopes for the NEM and $d_5$-NEM containing peptides. All missing ions were assigned a value equivalent to the minimum intensity, only unique peptides were quantified, and since this included quantitation at the peptide level, single channel was used. The H/L ratios of approximately 25% of the quantified peptides were manually validated.

## Bioinformatics analysis of cysteine-containing peptides

For functional annotation: Protein sequences from the FTP site of the Uniprot Protein database for mouse (Proteome_ID/Tax_ID: UP000000589/10090), rat (Proteome_ID/Tax_ID: UP000002494/10116), and human (Proteome_ID/Tax_ID: UP000005640/9606) release current as of May 23 2015. Sequence annotation in the feature fields (ACT_SITE, BINDING, CA_BIND, DISULFID, DNA_BIND, DOMAIN, METAL, MOD_RES, MOTIF, NP_BIND, SITE, ZN_FING) of the Uniprot entry was searched and any annotation corresponding to the labeled cysteine peptides was collected.

For redox cysteine annotation: each peptide identified by MS, all exact matches in any of the RedoxDB databases on any oxidative modification cysteine sequences (fasta or additional_fasta) were collected. For motif search: The lager data set of putative modification cysteine sites and their vicinity sequences were submitted to the pLogo program (www.plogo.uconn.edu, version v1.2.0) (*O'Shea et al., 2013*) to identify linear motif.

For prediction of candidate peptides for nitrosylation: the peptide sequences with H/L ratio >2 were submitted for use in predicting nitrosylation sites under the medium threshold condition using the batch prediction tool of the GPS-SNO 1.0 software (*Xue et al., 2010*). The predicted nitrosylation sites of sequences were extracted for further analysis.

For determination of surface accessibility and secondary structural motif: we turned to DSSP's (*Kabsch and Sander, 1983*; *Touw et al., 2015*) annotations of the PDB (*Berman et al., 2000*). We downloaded a total of 108355 DSSP-annotated PDB files from (rsync.cmbi.ru.nl/dssp/) on Sep. 9th 2015. Each peptide with H/L greater than 2-fold was aligned on all matching DSSP profiles, from which the 10-state structural context and accessibility were extracted. When an exact match is not found, then all matches with 1 mismatch are considered. The structural context of a peptide is defined as the context that reoccur most frequently among the hits, while the accessibility is the average across the hits, $\log_2$-normalized by the median of accessibility considering the amino acid type; positive $\log_2$ values means that the amino acid embedded in the 3D structure of the protein is more accessible that the mode (median).

For pathway annotation: the canonical pathways were scored based on the total sulfhydrated peptides and the peptides with H/L ratios greater than 2 by using DAVID (www.david.ncifcrf.gov) and Ingenuity Pathway Analysis (IPA, www.qiagen.com/ingenuity) programs. Statistical significant of pathways are calculated, and pathways are ranked by the p-values based on those tests. The tests measure the likelihood that the association between proteins measured in our experiments and a pathway is due to random chance. The smaller the p-value the less likely that the association is due to random chance. Top scoring pathways are presented.

## Metabolic labeling of MIN6 cells

MIN6 cells were plated onto 10-cm plates in triplicates and cultured in the cell growth medium. After 48 hr treatment with Tg, metabolic labeling was performed. Cells were washed with warm PBS and incubated for 3 hr in the DMEM medium containing 10% heat inactive FBS, 2 mM glutamine, and 25 mM glucose consisting of a mixture of 12.5 mM d-glucose plus 12.5 mM of d-[U-$^{13}$C] glucose. After incubation, cells were washed with PBS, followed by trypsinization. Cells were pelleted by centrifuging at 4°C for 5 min at 650 x *g*, and pellets were stored at -80°C until extraction of metabolites.

## Assay of media [U-$^{13}$C] glucose enrichment

Glucose isotopic enrichment was determined following (*van Dijk et al., 2001*) with modifications. Briefly, glucose was extracted by the addition of 500 µl of ice-cold ethanol to 50 µl of media. Samples were mixed and incubated on ice for 30 min. Samples were centrifuged at 4°C for 10 min at 14,000 rpm and ethanol was transferred to a GC/MS vial and evaporated to dryness in a SpeedVac evaporator. Glucose was converted to its pentaacetate derivative by the reaction with 150 µl of acetic anhydride in pyridine (2:1, v/v) at 60°C for 30 min. Samples were evaporated to dryness and glucose derivatives were reconstituted in 80 µl of ethyl acetate and transferred to GC/MS inserts. Samples were injected in duplicate and masses 331–337, containing M0...M+5 isotopomers were monitored. Enrichment was determined as a ratio of (M+5) / ($\Sigma_{M0-M5}$).

## Metabolite extraction

Metabolites were extracted following (*Yang et al., 2008*) with modifications. Briefly, cellular pellets in Eppendorf tubes were homogenized and frozen in 600 µl of Folch solution (chloroform:methanol, 2:1, vol./vol.) on dry ice. After addition of 0.4 volumes of ice-cold water, cells were homogenized again and let sit on ice for 30 min. Homogenates were centrifuged at 4°C for 10 min at 14,000 rpm. The upper methanol/water layer was removed to GC/MS vial. To the bottom chloroform layer, 120 µl of water and 200 µl of methanol were added and extraction steps from above were repeated. Combined methanol/water layers were evaporated to dryness in Speedvac evaporator at 4°C. Metabolites were derivatized using two-step derivatization. First, keto- and aldehyde groups were protected by the reaction with MOX (methoxyamine-HCl in pyridine, 1:2) overnight at room temperature. Then excess derivatizing agent was evaporated and dry residue was converted to MOX-TMS (trimethylsilyl) derivative by reacting with bis(trimethylsilyl) trifluoroacetamide with 10% trimethylchlorosilane (Regisil) at 60°C for 20 min. Resulting MOX-TMS derivatives were run in GC-MS.

## GC-MS conditions

Analyses were carried out on an Agilent 5973 mass spectrometer equipped with 6890 Gas Chromatograph. A DB17-MS capillary column (30 m × 0.25 mm × 0.25 µm) was used in all assays with a helium flow of 1 ml/min. Samples were analyzed in Selected Ion Monitoring (SIM) mode using

electron impact ionization (EI). Ion dwell time was set to 10 ms. The following metabolites were monitored: Glycerol 3 phosphate (G3P), 3 Phosphoglycerate (3PG), Lactate, Alanine, Citrate, Succinate, Fumarate, Malate, Glycine, and Serine.

## Mitochondrial oxidative phosphorylation and glycolysis

MIN6 cells were diluted to 80 000 cells/well in a Seahorse tissue culture system in the presence of either *GFP* or *ATF4* overexpression. Cells were plated 2 days prior to experimentation. The cells were washed with warm PBS and then incubated for 30 min at 37°C and ambient $CO_2$ in $HCO_3$-free DMEM containing 25 mM glucose, 2 mM glutamine, 1 mM pyruvate (pH 7.4). Cells were then treated sequentially with oligomycin (0.2 µg/ml), carbonyl cyanide 4-(trifluoromethoxy) phenylhydrazone (FCCP, 1 µM) and rotenone (1 µM). The rates of mitochondrial respiration and cellular acidification were determined by using the Seahorse extracellular flux analyzer (Seahorse Bioscience, North Billerica, MA). Corrected oxygen consumption rate (OCR) and extracellular acidification rate (ECAR) values were normalized to cell number.

## Other methods

MIN6 cells extracts for protein immunodetection were obtained after cell lysis in RIPA buffer as described before (*Krokowski et al., 2013*). Protein content was determined by the BCA assay (Bio-Rad). Mouse islet protein extracts were extracted in RIPA buffer. From approximately 100 islets the same amount of extracts determined by measuring DNA content with the Quant-iT dsDNA assay kit (Molecular Probes) and equal DNA amount was used for immunodetection. Western blotting was performed as described before (*Krokowski et al., 2013*). Anti-Actin (ab 3280) antibodies were from Abcam. Anti-CTH (H00001491) and anti-CBS (H00000875) were from Abnova. Antibodies against: PERK (3192), phospho PERK (3179) and PKM2 (4053) were purchased from Cell Signaling. Anti-ATF4 (sc-200), anti-GAPDH (sc-32233), anti-eIF2α (sc-133227), and XBP1 (sc-7160) were from Santa Cruz Biotechnology. Antibodies against phosphorylated eIF2α (NB 110–56949) and Ero1α (NB 100–2525) were obtained from Novus and anti-tubulin (T9026) serum was from Sigma-Aldrich.

## Acknowledgements

We thank Dr. Jonathan S Stamler for providing valuable comments in the development of methodology. Dr. John J Mieyal provided suggestions for GAPDH glutathionylation study. Bernard Tandler provided editorial assistance. We thank Dr. Solomon Snyder for providing us with plasmids for recombinant GAPDH expression. We thank Dr. Martin Jendrisak and the entire team of the Gift of Hope Organ and Tissue Donor Network in Chicago for the human pancreas tissues used in this study. We also thank Jing Wu, Xia Liu, Scott A. Becka, Case Western Reserve University, for technical assistance. We thank Ken Farabough for assistance with the manuscript preparation. We acknowledge the Vector Biolabs (Philadelphia, USA) providing the adenovirus-mediated shRNA for CTH knockdown experiment. This work was supported by NIH grants R37-DK060596, R01-DK053307 (to MH), R01-HL58984 (to RB), R01-DK013499 (to ML) and NIH DK48280 (to PA), Canada Excellence Research Chair grant CERC08 (to LD), and an American Heart Association grant (13SDG17070096 to O.K.). The Orbitrap Elite instrument was purchased via an NIH shared instrument grant 1S10RR031537-01.

## Additional information

### Funding

| Funder | Grant reference number | Author |
| --- | --- | --- |
| National Institutes of Health | R01-DK053307 | Xing-Huang Gao<br>Dawid Krokowski<br>Bo-Jhih Guan<br>Mithu Majumder<br>Maria Hatzoglou |
| Canada Excellence Research Chairs, Government of Canada | CERC08 | Marc Parisien<br>Luda Diatchenko |
| American Heart Association | 13SDG17070096 | Omer Kabil |

| National Institutes of Health | R37-DK060596 | Xing-Huang Gao<br>Dawid Krokowski<br>Bo-Jhih Guan<br>Mithu Majumder<br>Maria Hatzoglou |
| --- | --- | --- |
| National Institutes of Health | R01-HL58984 | Ruma Banerjee |
| National Institutes of Health | R01-DK013499 | Ming Liu |
| National Institutes of Health | DK48280 | Peter Arvan |
| National Institutes of Health | 1S10RR031537-01 | Belinda Willard |

The funders had no role in study design, data collection and interpretation, or the decision to submit the work for publication.

## Author contributions

XHG, Conceived the study; Designed and performed the experiments; Prepared the manuscript and analysed and interpreted the data; DK, Contibuted expertise on the stress response and amino acid uptake studies; Additionally, he has actively participated in the experimental design and data interpretation during the entire study; BJG, Contributed RNA analysis and manuscript preparation; IB, Performed and analysed the metabolic flux studies; MM, Contributed RNA analysis; MP, LD, Performed bioinformatics tasks and assisted in data interpretation; OK, Performed the H2S measurements; BW, Performed the proteomics analysis with LC-MS/MS; RB, Advised on H2S signaling and the chemistry of modifications of cysteine residues; Manuscript editing; BeW, Participated in the proteomics analysis of GAPDH and initial studies of the identification of the sulfhydrated proteome; GB, Performed the IPA analysis; CRE, Analyzed the Metabolomics data; PLF, Advised on cysteine modifications by nitrosylation; SLG, Assisted with the Seahorse analyzer; CLH, Provided expertise on cellular metabolism; Mitochondrial function and metabolic data analysis; ML, Provided expertise on β cell biology and data analysis concerning β cell function.; PA, Provided expertise on β cell biology and data analysis concerning β cell function; MH, Conceived the study; Designed the experiments; Prepared the manuscript and analysed and interpreted the data

## Ethics

Human subjects: Human Islet Study-Institutional review board approval for research use of isolated human islets was obtained from the University of Michigan (IRB number 2014-0069). Human islets were isolated from previously healthy, nondiabetic organ donors by the University of Chicago Transplant Center. Three independent human islet batches from two male donors aged 20 and 58 and one female donor aged 48 were used in this study.

Animal experimentation: Experimental protocols were approved by the Case Western Reserve University Institutional Animal Care and Use Committee.

## Additional files

### Supplementary files

• Supplementary file 1. Metabolite flux and relative concentrations of metabolites from control (CON) and Tg-treated MIN6 cells for 18 hr. PAG was added for the last 3.5 hr of Tg-treatment in the indicated experimental samples. Notes: the reported serine M+2 labeling reflects the two carbon atoms in the GC-MS fragment ion that was quantitated, and is expected to be predominantly derived from fully-labeled (M+3) serine.

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
