## [Decision Letter]

Thank you for submitting your work entitled "Quantitative H_2_S-mediated protein sulfhydration reveals metabolic reprogramming during the Integrated Stress Response" for peer review at *eLife*. Your submission has been favorably evaluated by Jim Kadonaga (Senior editor), a Reviewing editor, and two reviewers.

The reviewers have discussed the reviews with one another and the Reviewing editor has drafted this decision to help you prepare a revised submission.

Summary

*Reviewer 1*: This work is one of the first studies to analyze sulfhydration using a proteomic approach. A few points need to be addressed before the manuscript is suitable for publication in *eLife*. Overall, this is a novel, nicely done study.

*Reviewer 2*: The regulation of GAPDH via sulfhydration of Cys^150^ is convincing and supported by several lines of evidence. Using this protein as a model, the authors reasoned that other enzymes of central metabolism may be affected by ER stress. This approach was nicely extended to ATF4, and the data also looks convincing. Finally, a proteomic approach was developed resulting in 800 proteins targeted by sulfhydration. Overall, this is a novel, nicely done study

Major comments for revision:

1) *Reviewer 2*: A more thorough analysis of the resulting dataset of sulfhydrated proteins would be useful. First, a comment on controls is needed. Proteomics experiments are notorious for false positives. For example, two-thirds of all proteomics experiments, regardless of the design and the hypothesis tested, seem to identify GAPDH and some other proteins with redox Cys, such as peroxiredoxins. Appropriate controls could then be cells with inactive ATF4 or a parallel analysis of another redox modification. Another control could involve comparison of the dataset with CompPASS or other databases, which incorporate numerous experiments against which new datasets can be evaluated. Second, the dataset can be better analyzed for sequence and structural motifs and patterns, surface exposure, and protein abundance (more abundant proteins are more likely to be detected and are more likely to be involved in central metabolic processes). Gene enrichment analyses may be misleading and therefore require better statistical treatment.

2) *Reviewer 1*: Figure 1: The authors show the time course of induction of CSE and H_2_S production in response to thapsigargin. The levels of ROS and total GSH are measured too. The levels of ROS seem to be maximal at 12h. What about the GSH levels? The GSH levels are shown for the 3h time point and then at 18h? What are the levels of GSH at the 12h time point? From the experiments shown, there appears to be a distinct possibility that oxidative stress could contribute to the induction of CSE. Does the presence of an antioxidant blunt the induction of CSE and therefore H_2_S production in response to Tg?

3) *Reviewer 1*: Figure 2: ATF4-deficient cells exhibit an increase in sulfhydrated GAPDH, which is attributed to a possible increase in activity of CBS. Would an inhibitor of CBS or knockdown of CBS prevent this increase?

4) *Reviewer 1*: Can the authors comment on the time course of sulfhydration during ER stress? H_2_S production has been measured at different time points. A time course of sulfhydration using one protein as an example (such as GAPDH) should be shown using the biotin thiol assay or maleimide assay.

Other comments for consideration during revision:

Reviewer 1:

Figure 1: The activity of GAPDH decreases in the presence of the oxidant H_2_O_2_and NaHS prevents this inactivation in vitro. Can the authors demonstrate this protection in MIN6 cells with H_2_O_2_treatment?

Figure 2: Overexpression of ATF4 causes the upregulation of genes involved in cysteine disposition. Does Tg treatment also cause an increase in the uptake of cystine similar to overexpression of ATF4?

The authors identify several proteins which undergo sulfhydration during ER stress. Some cysteine residues on these proteins appear more reactive than others. Is there a relationship between the p*K*a of these cysteine residues and its susceptibility to sulfhydration as reported in literature. Could the authors list a few examples from the large data sets generated?

The authors should provide the list of proteins sulfhydrated under ER stress. It is mentioned in the Results and Discussion that over 834 cysteine-containing peptides were identified, of which 771 displayed average H/L ratio and 348 exhibited high H/L ratio. Only a few proteins are listed in the study.

Does Tg treatment for 18h cause a change in cell proliferation?

Can the authors comment on the time course of sulfhydration during ER stress? H_2_S production has been measured at different time points. A time course of sulfhydration using one protein as an example (such as GAPDH) should be shown using the biotin thiol assay or maleimide assay.

The authors should discuss the physiological consequences of increased glycolytic activity under ER stress.

Figure 1 shows DTT treatment. This is described in the legends, but not in the main text.

Subsection “Human Islets RNA isolation”: Were human islets treated with 1 mM? Did the authors mean 1 µM?

Subsection “Purification of H_2_S-modifieded cysteine-containing peptides from MIN6 cells”. Correct spelling to H_2_S-modified

Reviewer 2:

1) The Introduction should be improved. A direct oxidation of Cys to form sulfenylated species is not mentioned, though it is stated that the oxidized sulfur atom may react with sulfide to yield sulfhydration. Overall, both the rationale for the study and the literature review may be improved.

2) The use of light/heavy NEM is not well explained. Is the purpose to quantify reactivity, similar to the approach of Weerapana et al. Nature 2010 (21085121)? Then, the light/heavy NEM should be used in different ratios. The way the method is presented is confusing.

3) The new proteomics method developed assumes that sulfhydration is the only Cys modification that is targeted by the method. However, any dithiol protein/peptide/compound would give the same profile of eluted target peptides. This should be acknowledged and discussed, as it is possible that some of the target proteins identified in the study have other modifications.

---

## [Author Response]

*Major comments for revision: 1)* Reviewer 2*: A more thorough analysis of the resulting dataset of sulfhydrated proteins would be useful. First, a comment on controls is needed. Proteomics experiments are notorious for false positives. For example, two-thirds of all proteomics experiments, regardless of the design and the hypothesis tested, seem to identify GAPDH and some other proteins with redox Cys, such as peroxiredoxins. Appropriate controls could then be cells with inactive ATF4 or a parallel analysis of another redox modification. Another control could involve comparison of the dataset with CompPASS or other databases, which incorporate numerous experiments against which new datasets can be evaluated. Second, the dataset can be better analyzed for sequence and structural motifs and patterns, surface exposure, and protein abundance (more abundant proteins are more likely to be detected and are more likely to be involved in central metabolic processes). Gene enrichment analyses may be misleading and therefore require better statistical treatment.*

1) In this study, we have evaluated and compared our datasets with several databases as follows: Uniprot, NCBI, PDB, RedoxDB and GPS-SNO. Four bioinformatics programs were employed for this purpose, including pLOG sequence motif analyzer, DSSP-PDB structure analyzer, DAVID and IPA pathway annotations.

2) It has been reported that iNOS expression is induced by ER stress, which may contribute to protein nitrosylation. GAPDH is one of the nitrosylated targets under nitrosative stress. We have thus examined if iNOS protein expression is induced in MIN6 cells treated with Tg, using Western blot analysis. We found that iNOS protein was not induced. Furthermore, we have performed the BTA and BST (Biotin Switch Technique) assays in parallel on the same lysate extracted from MIN6 cells treated with Tg for 18h in order to identify if NO nitrosylates GAPDH. We found an increase in GAPDH sulfhydration and not nitrosylation. It is known that the GAPDH activity is inhibited by nitrosylation or glutathionylation or other redox-based modifications. However, our data showed that GAPDH activity increases by ER stress. All together, our data support that sulfhydration and not other redox-based cysteine modifications, primarily occur on proteins in MIN6 cells during the time frame of chronic ER stress that was studied in this manuscript.

3) We performed the sequence alignment analysis of the quantified peptide data using a sequence annotation program named pLOG at http://plogo.uconn.edu/. Similar to ComPASS, the pLOG program has been broadly employed and has been cited in 30 articles until now. It does not only provide real-time conditional probability calculations for motif assessments, but also presents specific features for the analysis of post-translational modifications (nitrosylation) based on peptide sequences (1). By using the pLOG, we showed that an Arg residue was significantly enriched at the +1 position of the modified cysteine when we selected only the peptides with high H/L ratios (2-fold), but no conserved residues were found to surround the modified cysteine residue based on the full pair-labeled peptides. These data were presented in Figure 3—figure supplement 4 and discussed in the Results (paragraph thirteen).

The surface accessibility and structure motif analysis were performed by using the DSSP program. We downloaded a total of 108355 DSSP-annotated PDB files from rsync.cmbi.ru.nl/dssp/. Each peptide with H/L greater than 2-fold was aligned on all matching DSSP profiles, from which the 10-state structural context and accessibility were extracted. The analysis showed that the modified cysteine residue is highly accessible and preferentially located at the N-terminal of alpha helix of the protein. These data were included in Figure 3—figure supplement 4 and discussed the aforementioned paragraph of the Results section.

We have performed an experiment in order to examine if there is a correlation between protein sulfhydration and their protein abundance. We chose to identify the entire proteome and sulfhydrome in Min6 cells treated with Tg for 18h because we would be able to better understand the metabolic response of the cells described in our manuscript. We therefore identified almost 1000 cysteine sulfhydration sites in 868 proteins by the BTA approach, and quantified the full proteome from MIN6 cells treated the Tg by using label-free, semi-quantitative MS approaches. Then, we compared the full protein abundance with two sulfhydrome datasets from either the Tg-treated, or ATF4 overexpressing MIN6 cells. The analysis showed that, protein sulfhydration does not correlate with their protein abundance. The data were presented in Figure 3—figure supplement 6, Figure 3—figure supplement 7 and discussed the Results (paragraph ten).

4) We have employed the second gene annotation program IPA (Ingenuity Pathway Analysis, http://www.ingenuity.com/products/ipa) to calculate the pathway enrichment of the quantified peptides. Similar to the results generated by DAVID pathway annotation program, the IPA analysis showed that glycolysis is among the top 5 highest enriched pathways based on the peptides with enriched H/L ratios (>2-fold), confirming that glycolytic enzymes are primarily targeted by increased H_2_S levels in MIN6 cells. These data are presented in Figure 3—figure supplement 5, and discussed the Results (paragraph thirteen).

*2)* Reviewer 1*: Figure 1: The authors show the time course of induction of CSE and H_2_S production in response to thapsigargin. The levels of ROS and total GSH are measured too. The levels of ROS seem to be maximal at 12h. What about the GSH levels? The GSH levels are shown for the 3h time point and then at 18h? What are the levels of GSH at the 12h time point? From the experiments shown, there appears to be a distinct possibility that oxidative stress could contribute to the induction of CSE. Does the presence of an antioxidant blunt the induction of CSE and therefore H_2_S production in response to Tg?*

1) We performed an experiment to quantify GSH levels and GSH/GSSG ratios during Tg treatment of Min6 cells, including the time points suggested by the reviewer. It showed that the GSH/GSSG ratio diminished by 50% at 12h of Tg treatment compared to untreated cells, despite the total levels of GSH that increased significantly starting at 12h of Tg treatment. These data indicate that severe oxidative stress occurs during chronic ER stress. The new data are presented in Figure 1.

2) We performed an experiment to test if the induction of CSE levels was affected by antioxidant treatments in MIN6 cells treated with Tg for 18h. N-acetylcysteine (NAC) and catalase (CAT) were employed as antioxidant scavengers for this purpose. The experiment showed that the antioxidants did not change the levels of induction of the CSE protein, but had protective effects on induction of apoptosis. These data are not shown in the manuscript. We chose not to include the data in the manuscript, because we plan to expand these studies and determine the significance of CSE-mediated H_2_S production during ER stress on cell fate.

Author response image 1.Antioxidant treatments do not affect the induction of CES protein expression in MIN6 treated with Tg for 18h.MIN6 cells were either kept in DMEM growth medium (10% heat inactive FBS, 2 mM glutamine, and 25 mM glucose) as control, or were cultured in the medium supplement with Tg alone, Tg and N-acetylcysteine (NAC, 1 mM), or Tg and catalase (CAT, 500 unit/mL) for 18h. After treatments, cells were washed with cold PBS and lysed with the RIPA buffer (150 mM NaCl, 1 mM EDTA, 0.5% Triton X-100, 0.5% deoxycholic acid and 100 mM Tris-HCl (pH 7.5) containing protease and phosphatase inhibitor from Roche. Cells were then sonicated on ice, lysates were clarified by centrifugation at 4°C and the protein concentrations were determined by the BCA assay (BioRad). (Top panel) The antioxidant effects were evaluated by the caspase 3 activity assay. We found that the caspase 3 activation was significantly reduced by both antioxidant treatments compared to untreated cells. However, the induction of CSE protein expression was unaffected by the treatments (bottom panel), indicating that Increased ROS production does not contribute in induction of CSE protein levels via the ATF4 transcription program during ER stress in MIN6.**DOI:**
http://dx.doi.org/10.7554/eLife.10067.037

*3)* Reviewer 1*: Figure 2: ATF4-deficient cells exhibit an increase in sulfhydrated GAPDH, which is attributed to a possible increase in activity of CBS. Would an inhibitor of CBS or knockdown of CBS prevent this increase?*

We have used the CBS inbibitor AOAA (aminooxyacetic acid), and showed that the basal sulfhydration of GAPDH is abolished. This suggested that CBS controls protein sulfhydration during basal non-stress conditions. The data in our manuscript also suggest that CSE induction during ER stress further increases the sulfhydration of proteins. We plan to study the regulation of CBS in the absence of stress in Min6 cells which are deficient of UPR markers (ATF4, PERK, XBP1, other) and further understand how stress influences Min6 metabolism and insulin secretion (GSIS). We hope the reviewers can allow this. On the other hand, we have knocked down CSE levels via the use of a shRNA, in MIN6 cells treated with Tg, and showed that protein sulfhydration decreased. Two targets of sulfhydration, GAPDH and PHGDH (phosphoglycerate dehydrogenase), were shown to have decreased levels of sulfhydration. The data were included in Figure 2—figure supplement 11, and discussed in paragraphs ten and eleven of the Results.

*4)* Reviewer 1*: Can the authors comment on the time course of sulfhydration during ER stress? H_2_S production has been measured at different time points. A time course of sulfhydration using one protein as an example (such as GAPDH) should be shown using the biotin thiol assay or maleimide assay.*

We performed the BTA for GAPDH and PHGDH, two proteins identified as being sulfhydrated. The peak of sulfhydration was observed at 12 hours, which was also the peak of ROS levels. Nevertheless, at the 18th hour, the sulfhydration was significantly higher than basal levels. These data are now included in Figure 2—figure supplement 10 and discussed in the Results (paragraph ten).

*Other comments for consideration during revision:* Reviewer 1:

*Figure 1: The activity of GAPDH decreases in the presence of the oxidant H_2_O_2_ and NaHS prevents this inactivation in vitro. Can the authors demonstrate this protection in MIN6 cells with H_2_O_2_ treatment?*

We thank the reviewer for this suggestion. We want to perform such studies and determine the interplay between oxidants and H_2_S during chronic ER stress. As mentioned above (Response 2 to comment 2), we plan to study changes in the balance between oxidants and H_2_S on cell fate during chronic ER stress.

*Figure 2: Overexpression of ATF4 causes the upregulation of genes involved in cysteine disposition. Does Tg treatment also cause an increase in the uptake of cystine similar to overexpression of ATF4?*

As requested by the reviewer, we now show that Tg-treatment also increases the plasma membrane activity of glutamate/cystine flux. This was shown in Figure 1—figure supplement 4 and discussed in the Results (paragraph two).

*The authors identify several proteins which undergo sulfhydration during ER stress. Some cysteine residues on these proteins appear more reactive than others. Is there a relationship between the p*K*a of these cysteine residues and its susceptibility to sulfhydration as reported in literature. Could the authors list a few examples from the large data sets generated?*

As noted in the manuscript, a number of the proteins identified by the BTA method have been reported previously to be targets of sulfhydration, glutathionylation or nitrosylation, indicating the presence of reactive cysteines susceptible to modification. As discussed in the excellent perspective article below (2), while a lower p*K*a means higher availability of the thiolate at neutral pH, it is often associated with lower nucleophilicity. Hence a lower p*K*a does not necessarily agree with increased reactivity. Instead, the efficiency of modification is modulated by multiple factors including transition state stabilization and leaving group activation.

*The authors should provide the list of proteins sulfhydrated under ER stress. It is mentioned in the Results and Discussion that over 834 cysteine-containing peptides were identified, of which 771 displayed average H/L ratio and 348 exhibited high H/L ratio. Only a few proteins are listed in the study.*

Three datasets related to the manuscript are now presented as source files.

*Does Tg treatment for 18h cause a change in cell proliferation? Can the authors comment on the time course of sulfhydration during ER stress? H_2_S production has been measured at different time points. A time course of sulfhydration using one protein as an example (such as GAPDH) should be shown using the biotin thiol assay or maleimide assay.*

The reviewer is probably thinking that the increased glycolysis and decreased mitochondria function may be a similar response to the Warburg effect in cancer cells. It may be expected to increase glycolytic intermediates that lead to synthesis of biomass for proliferation/growth. Although very interesting, we feel that this is outside the scope of this manuscript. However, the reviewer’s question is valued and should be followed in future studies. It has been reported recently, that increased insulin synthesis in pancreatic β cells promotes proliferation in a manner that involves the induction of the Unfolded Protein Response (3). It is therefore possible that increased insulin synthesis in β cells, causes induction of the UPR that acts as a protective mechanism in the continuing insulin synthesis and therefore proliferation. In contrast to this response, β cells that do not induce the UPR, they cannot properly fold and handle the additional insulin and therefore cannot have positive effects of insulin on proliferation. Future studies can explore the involvement of H_2_S-metabolic responses in β cells during the pre-diabetes state that β cells increase insulin synthesis in response to peripheral tissue insulin resistance.

A detailed explanation on the second part of the reviewer’s comment is given in the response to comment 4 above.

*The authors should discuss the physiological consequences of increased glycolytic activity under ER stress.*

The physiological consequences of H_2_S production under ER stress in β cells are now discussed in detail (please see paragraphs one and four of the Discussion). We compared the β cell response to ER stress with the Warburg effect in tumor cells and explained on how the H_2_S-metabolic reprogramming can help β cell adaptation to stress, such as to sustain insulin synthesis and secretion.

Figure 1 shows DTT treatment. This is described in the legends, but not in the main text.

Done.

*Subsection “Human Islets RNA isolation”. Were human islets treated with 1 mM? Did the authors mean 1 µM?*

This has been corrected.

*Subsection “Purification of H_2_S-modifieded cysteine-containing peptides from MIN6 cells”. Correct spelling to H_2_S-modified*

Done.

Reviewer 2:

*1) The Introduction should be improved. A direct oxidation of Cys to form sulfenylated species is not mentioned, though it is stated that the oxidized sulfur atom may react with sulfide to yield sulfhydration. Overall, both the rationale for the study and the literature review may be improved.*

The Introduction section was improved and updated with additional references as requested by the reviewer.

*2) The use of light/heavy NEM is not well explained. Is the purpose to quantify reactivity, similar to the approach of Weerapana et al. Nature 2010 (21085121)? Then, the light/heavy NEM should be used in different ratios. The way the method is presented is confusing.*

A paragraph to explain this quantitative approach was added to the Results.

*3) The new proteomics method developed assumes that sulfhydration is the only Cys modification that is targeted by the method. However, any dithiol protein/peptide/compound would give the same profile of eluted target peptides. This should be acknowledged and discussed, as it is possible that some of the target proteins identified in the study have other modifications.*

If a protein contains other than sulfhydration, redox-based modifications on cysteine residues (disulfide, glutathionylated or nitrosylated) will not react with NM-biotin, therefore any protein with those types of modifications will not bind on the avidin column by the BTA. We have discussed this throughout the manuscript (please refer to the Results section, paragraphs nine and ten).